# Interface preassembly oriented growth strategy towards flexible crystalline covalent organic framework films for OLEDs

Xiang-Chun Li[1], Hao Sun[1], Zuqiang Wang[1], Weijie Yang[1], Qiaoyu Wang[1], Chuanrui Wu[1], Jiajun Chen[1], Qinchen Jiang[1], Ling-Jun He[1], Qian Xue[1], Wei Huang [1,2] & Wen-Yong Lai [1] ✉

The synthesis of flexible crystalline films for optoelectronic applications remains a significant chemical challenge due to the inherent contradiction between flexibility and crystallinity. The delicate balance between flexibility and crystallinity has long constituted a barrier to the development of high-performance optoelectronic materials. Herein, an interface preassembly oriented growth (IPOG) strategy has been explored to fabricate flexible crystalline covalent organic framework (COF) films with controllable thickness. By synergistically modulating hydrophilic and hydrophobic interactions along with interfacial confinement, a set of uniform and flexible crystalline COF films were successfully synthesized. This achievement unlocks the potential of COFs for device applications in organic light-emitting diodes, leading to unprecedented high-efficiency electroluminescence from COFs. This groundbreaking advancement not only lays the foundation for the progress of COF-based OLEDs but also signifies the advent of an era in the synthesis of flexible crystalline materials, wherein exceptional mechanical properties are seamlessly integrated with superior electronic performance, thus heralding a transformative impact on the landscape of flexible electronics.

The emerging field of covalent organic frameworks (COFs) has become a cornerstone in materials and chemistry, providing a versatile platform for the design of porous crystalline polymers with atomic precision[1–3]. COFs have been extensively studied for their application in gas separation[4], catalysis[5,6], energy storage[7,8], sensing[9], nanogenerators,[10] and photothermal therapy[11], with a growing emphasis on their potential application in emerging optoelectronic devices[12,13]. Unlocking the full potential of COFs for optoelectronic devices necessitates the integration of crystallinity and flexibility. However, a fundamental trade-off exists: increasing the crystallinity of COFs inevitably compromises their mechanical properties, resulting in materials that are stiffer and more brittle. The primary challenge lies in achieving a delicate balance among flexibility,

crystallinity, and even luminescence, where precise control of material growth is crucial. Recent advances have been achieved in the synthesis strategies of various COFs, paving the way for the optimization of flexibility, crystallinity, and enhanced optoelectronic performance[14–17]. Nature has harnessed hydrophilic-hydrophobic interactions to control the growth of diverse structures with remarkable precision. The delicate balance of hydrophilic and hydrophobic interactions serves as a fundamental principle governing the assembly of biological systems across scales. Inspired by these natural processes, exploring these interactions to direct the controlled growth of thin films in COF synthesis is essential for overcoming current limitations. Controlling the orientation and alignment of building blocks through hydrophilic-hydrophobic

[1]State Key Laboratory of Flexible Electronics (LoFE), Institute of Advanced Materials (IAM), School of Chemistry and Life Sciences, Nanjing University of Posts & Telecommunications, 9 Wenyuan Road, Nanjing 210023, China. [2]Frontiers Science Center for Flexible Electronics (FSCFE), MIIT Key Laboratory of Flexible Electronics (KLoFE), Northwestern Polytechnical University, Xi'an 710072, China. ✉e-mail: iamwylai@njupt.edu.cn

interactions presents a promising approach to overcoming the aforementioned challenges.

This study presents a method for the interface preassembly oriented growth (IPOG) of flexible crystalline COF thin films with controllable thickness, resulting in a significant enhancement of their luminescence. The luminescent properties, flexibility and crystallinity of COF films were systematically regulated by the strategic control of the hydrophilic and hydrophobic interactions. Amphiphilic and hydrophilic COF films exhibited enhanced luminescence efficiency and flexibility as compared to their non-substituted and hydrophobic counterparts. These amphiphilic crystalline COF films are not only flexible but also exhibit tensile strains of up to 8.9%. Notably, the photoluminescence quantum yield (PLQY) of **PhPeMa-TAPB-COF** was 60 times greater than that of **Ph2Pe-TAPB-COF** in solid film. Furthermore, in order to evaluate the potential of these luminescent COF films for high-performance OLEDs. The flexible OLEDs utilizing COFs as active layers have been successfully constructed. The successful integration of flexibility and crystallinity in COF-based materials represents a significant advancement, opening up the possibilities for the future generation of flexible crystalline materials to combine excellent mechanical flexibility with high optoelectronic performance for flexible electronics.

## Results and discussion

The synthetic routes for the hydrophobic monomer (**Ph2Pe-2CHO**), amphipathic monomer (**PhPeMa-2CHO**), and hydrophilic monomer (**Ph2Ma-2CHO**) are shown in Supplementary Fig. S1, and the linkers were characterized via [1]H NMR, [13]C NMR and mass spectrum (Supplementary Fig. S2-S10). In Fig. 1a, an IPOG method was developed to induce the formation of an ordered preassembly structure at the interface of the organic and aqueous phases by adjusting hydrophilic and hydrophobic interactions. Subsequently, free-standing flexible crystalline COF films with good luminescence were prepared through acetic acid-catalyzed liquid-liquid interfacial polymerization at room temperature (RT). In liquid-liquid interfacial polymerization, the significance of the hydrophilic/hydrophobic properties of the reaction monomers is often underestimated, resulting in monomer diffusion and reaction in the oil and water phases, thereby affecting the quality and crystallinity of the resulting COF films[18–20]. By introducing hydrophilic/hydrophobic chains into the monomers, self-assembly at the liquid-liquid interface restrains the ordered polymerization of the monomers. Hydrophobic **Ph2Pe-TAPB-COF**, amphiphilic **PhPeMa-TAPB-COF**, and hydrophilic **Ph2Ma-TAPB-COF** were synthesized to explore the impact of hydrophilic/hydrophobic chains on film formation (Fig. 1b). Through optimization of the preparation conditions, high-crystallinity and excellent flexibility were achieved for amphiphilic and hydrophilic COF films with controllable thickness. The crystalline nature of **Ph2Pe-TAPB-COF, PhPeMa-TAPB-COF**, and **Ph2Ma-TAPB-COF** was confirmed via X-ray diffraction (XRD) (Fig. 1c), exhibiting intense diffraction peaks at 2.79°, 2.74°, and 2.72°, respectively, indicating high crystallinity. The Pawley-refined XRD patterns with favorable residuals for **Ph2Pe-TAPB-COF** ($R_p = 8.48\%$, $R_{wp} = 15.25\%$) and **Ph2Ma-TAPB-COF** ($R_p = 7.93\%$, $R_{wp} = 14.02\%$) align well with the experimental patterns. The smaller difference in Pawley-refined XRD patterns for amphiphilic **PhPeMa-TAPB-COF** (4.12%, $R_{wp} = 5.61\%$) suggests that amphiphilic chains promote COF crystallization more effectively. The nanocrystalline powders formed after grinding or ultrasonic dispersion of **PhPeMa-TAPB-COF** films show similar crystallinity to that of COF films, indicating that the COF films have good crystal stability (Supplementary Fig. S11). Compared with **Ph2Pe-TAPB-COF, PhPeMa-TAPB-COF**, and **Ph2Ma-TAPB-COF**, non-substituted analogs **Ph-TAPB-COF** (Supplementary Fig. S12) was synthesized. Pawley refined XRD patterns of **Ph-TAPB-COF** with relatively big difference (residuals $R_p = 11.65\%$, $R_{wp} = 16.05\%$), indicating that non-substituted linkers were not conducive to COF crystallization

(Supplementary Fig. S13). To validate the universality of this synthesis method, four types of amphiphilic COFs (**PhPeMa-TTr-COF, PhPeMa-TPA-COF, PhPeMa-TOB-COF**, and **PhPeMa-Py-COF**, Supplementary Fig. S14) and four types of hydrophilic COFs (**Ph2Ma-TTr-COF, Ph2Ma-Py-COF, Ph2Ma-TOB-COF**, and **3D-Ph2Ma-SpF-COF**, Supplementary Fig. S15) with different topologies were designed and synthesized using IPOG strategy. All amphiphilic and hydrophilic COF films exhibited strong diffraction peaks and certain crystallinity, with refined XRD patterns consistent with experimental profiles (Supplementary Fig. S16-S23).

The presence of rotationally labile imine bonds and the π-π stacking interactions between the layered conjugated structures can lead to significant emission quenching, attributed to thermal dissipation and aggregation-caused quenching[21]. The low luminescent characteristics and poor film-forming properties of imine-bonded COF hinder their practical application as emissive materials for optoelectronic devices. IPOG strategy has been developed to enhance the luminescent qualities of imine-bonded COFs by integrating different side chains into the linker. This approach effectively suppresses both intramolecular rotation and interlayer π-π interactions (Fig. 2a). As shown in Fig. 2b and Supplementary Fig. S24, the films of **Ph2Pe-TAPB-COF, PhPeMa-TAPB-COF**, and **Ph2Ma-TAPB-COF** exhibit continuous and uniform thin films with dimensions of ~20 cm². Under an optical microscope (Supplementary Fig. S25), the observed area appears to have no visible defects or fractures, except for a few wrinkles that are expected in the manipulation of thin films. In general, the hydrophilic and hydrophobic substituted COF films exhibit high integrity and smoothness on a macroscopic scale. Conversely, the film of the non-substituted analogs, **Ph-TAPB-COF**, displays numerous cracks, indicative of poor film formation. Notably, through the precise control of monomer concentrations and reaction times, the COFs can generate uniform free-standing films with controllable thickness. It was observed that the thickness of the COF films increased in response to elevated monomer concentrations and extended reaction times (Supplementary Fig. S26). Specifically, the thickness of the free-standing films can be effectively regulated by adjusting the monomer concentration. At a constant reaction time, an increase in monomer concentration resulted in a gradual transition of the film from transparent to opaque, indicating a corresponding increase in thickness. Scanning electron microscopy (SEM) was utilized for a more quantitative analysis, confirming a flat surface morphology. Top-view and cross-sectional SEM images illustrate the integrity and continuity of the COF films (Fig. 2c and Supplementary Fig. S27). Cross-sectional views revealed film thicknesses ranging from 100–300 nm (Supplementary Fig. S28), with the ability to adjust film thickness from nanometers to micrometers by modifying monomer concentrations and reaction times. Thus, the IPOG strategy facilitates the production of homogeneous films with considerable areas (around 20 cm²), representing an aspect ratio of 5 orders of magnitude between thickness and length scales. The surface properties of COF films were further characterized using atomic force microscopy (AFM) analysis (Supplementary Fig. S29). The results indicate that the amphiphilic **PhPeMa-TAPB-COF** films exhibit reduced roughness compared to the **Ph2Pe-TAPB-COF** and **Ph2Ma-TAPB-COF** films. This observation suggests that the introduction of amphiphilic chains facilitates the formation of a more uniform film morphology. Furthermore, the hydrophobicity of the synthesized COF films was assessed through water contact angle (CA) measurements. With increasing hydrophilic chains, the contact angle diminishes gradually, indicative of enhanced hydrophilicity (Supplementary Fig. S30). The different wetting behavior predominantly arises from the presence of hydrophilic and hydrophobic chains within the films. Amphiphilic **PhPeMa-TAPB-COF** films exhibit a significant difference in contact angle between water and oil surfaces, further demonstrating that the incorporation of hydrophilic and hydrophobic chains enhances the preassembly at the water-oil interface.

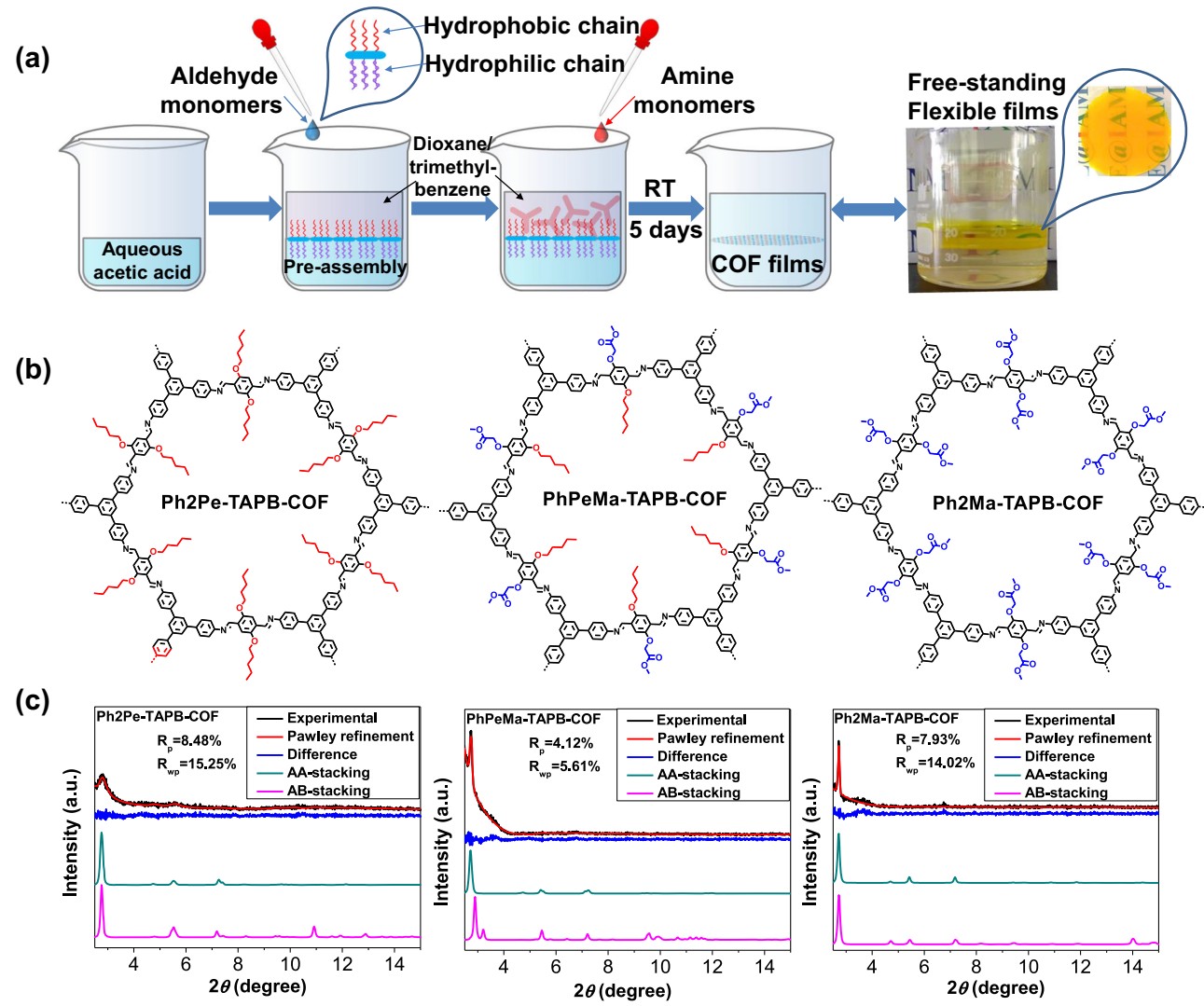

**Fig. 1 | Synthesis and COF films using IPOG strategy. a** Synthetic scheme for the preparation of the free-standing flexible COF films. **b** The chemical structure of **Ph2Pe-TAPB-COF**, **PhPeMa-TAPB-COF**, and **Ph2Ma-TAPB-COF**. **c** XRD patterns of the COF films and the simulated patterns: AA stacking mode (dark cyan) and AB eclipsed stacking mode (magenta) with the experimental pattern (black) combined Pawley refined pattern (red) and the difference (blue).

Interestingly, the introduction of hydrophilic and hydrophobic chains not only enhances the properties of COF films but also improves their luminescence characteristics. Upon exposure to 365 nm UV light, free-standing films of **PhPeMa-TAPB-COF** and **Ph2Ma-TAPB-COF** exhibited strong red fluorescence emission, in contrast to the absence of fluorescence in **Ph-TAPB-COF** and **Ph2Pe-TAPB-COF** (Fig. 2b and Supplementary Fig. S24). The photophysical properties of **Ph2Pe-TAPB-COF, PhPeMa-TAPB-COF**, and **Ph2Ma-TAPB-COF** films were systematically investigated using ultraviolet-visible (UV-vis) and photoluminescence (PL) spectroscopy. UV-vis spectra reveal intense absorption bands between 290 nm due to the π-conjugated **TAPB** group (Supplementary Fig. S31), with **PhPeMa-TAPB-COF** exhibiting stronger absorption within the 350–450 nm range as compared to **Ph2Pe-TAPB-COF** and **Ph2Ma-TAPB-COF**. Notably, the introduction of diverse hydrophilic and hydrophobic chains results in substantial variations in luminescent intensity (Fig. 2d). Compared to the hydrophobic **Ph2Pe-TAPB-COF** and hydrophilic **Ph2Ma-TAPB-COF**, the amphiphilic **PhPeMa-TAPB-COF** exhibits stronger fluorescence emission. PLQY of **PhPeMa-TAPB-COF** (2.38%) is 60 times greater than that of **Ph2Pe-TAPB-COF** (0.04%) in solid film (Supplementary Table S1). The layer distance of non-substituted analogs, **Ph-TAPB-COF**, measures 3.52 Å, while the layer distance of amphiphilic

**PhPeMa-TAPB-COF** extends to 4.01 Å (Fig. 2e). The moderate increase in COF layer distance can be attributed to steric repulsion from hydrophilic and hydrophobic chains. The incorporation of hydrophilic and hydrophobic chains limits the rotational movements of labile imine bonds and inhibits non-radiative transitions. Furthermore, the presence of hydrophilic and hydrophobic chains does not disrupt the interlayer interaction of COFs, aiding in enhancing charge carrier transport. To assess the porosity, $N_2$ adsorption analyses of the COF thin films were conducted at 77 K. The Brunauer-Emmett-Teller (BET) surface areas calculated for the three thin films were found to be 44 m²/g for **Ph2Pe-TAPB-COF**, 14 m²/g for **PhPeMa-TAPB-COF**, and 25 m²/g for **Ph2Ma-TAPB-COF** (Supplementary Fig. S32). The relatively low specific surface areas suggest that the introduction of hydrophilic and hydrophobic groups in these COFs results in the formation of dense films, thereby enhancing their flexibility. Additionally, the calculations using the Barret-Joyner-Halenda (BJH) method indicate narrow pore size distributions, with pore diameters of 3.7 nm for **Ph2Pe-TAPB-COF**, 3.4 nm for **PhPeMa-TAPB-COF**, and 3.3 nm for **Ph2Ma-TAPB-COF** thin films (Supplementary Fig. S33), which align well with the theoretically predicted pore sizes (about 3.1 nm). Thermogravimetric analysis indicates no weight loss for **Ph2Pe-TAPB-COF, PhPeMa-TAPB-COF**, and **Ph2Ma-TAPB-COF** until decomposition at

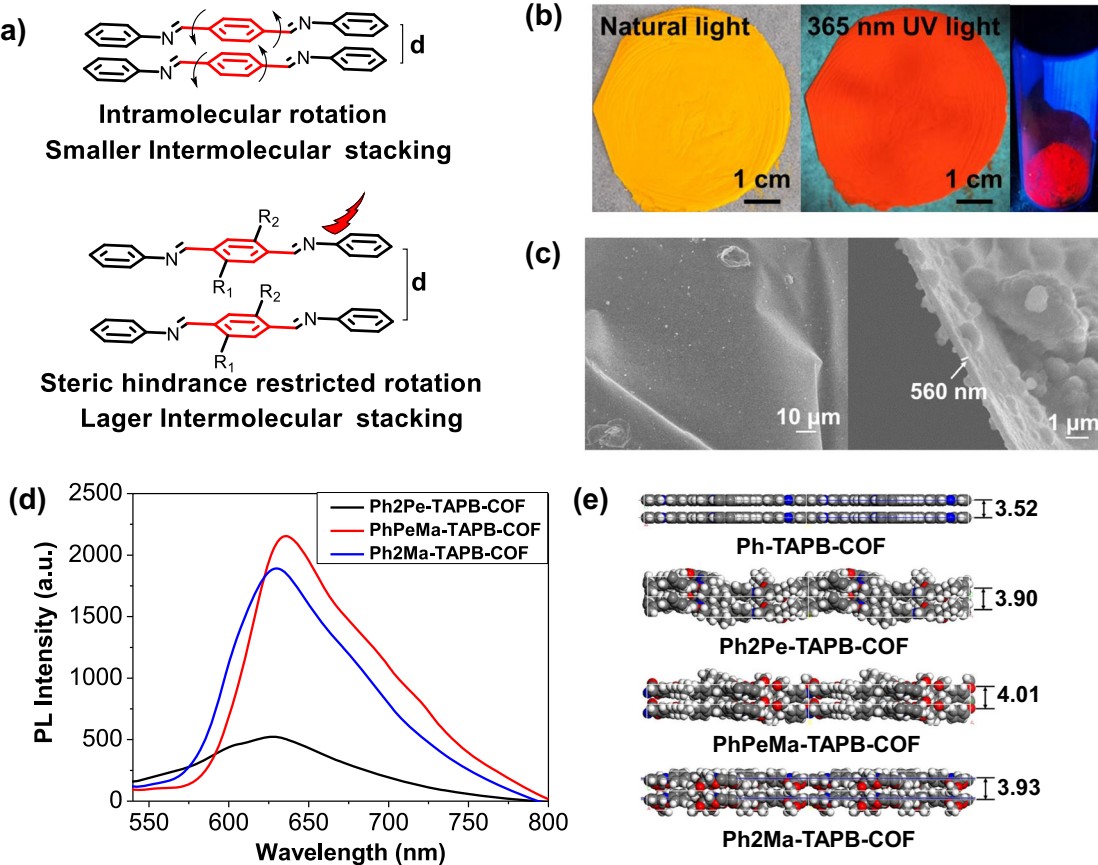

Fig. 2 | Photophysical properties of COF films. a Schematic diagram of imine-bonded COFs quenching emission and steric enhancing emission. b Photographs of **PhPeMa-TAPB-COF** film and red emission under 365 nm UV light. c Top-view (left) and cross-sectional view (right) SEM images of **PhPeMa-TAPB-COF** film. d Normalized PL spectra of **Ph2Pe-TAPB-COF, PhPeMa-TAPB-COF**, and **Ph2Ma-TAPB-COF** films. e The optimized crystal structures from side view of **Ph-TAPB-COF, Ph2Pe-TAPB-COF, PhPeMa-TAPB-COF**, and **Ph2Ma-TAPB-COF**.

280 °C, demonstrating high thermal stability (Supplementary Fig. S34). The impressive luminescent qualities, film-forming properties, and thermal stability of the COF films position them as promising candidates for utilization in electroluminescent devices.

To demonstrate the universality of the IPOG strategy, a series of reactions were conducted using various amine and aldehyde monomers to synthesize additional flexible crystalline COF films. Eight additional flexible crystalline two- and three-dimensional COF films were synthesized. As depicted in Fig. 3a and Fig. 3b, all amphiphilic COFs exhibited exceptional film-forming properties and strong luminescent characteristics, indicating that the luminescence enhancement induced by amphiphilic chains is a universally applicable method, regardless of the topology and core skeleton of COFs. In comparison to the amphiphilic COFs, the hydrophilic COFs also demonstrated good film formation and luminescent qualities (Fig. 3c and Fig. 3d). This amphiphilic IPOG strategy is not restricted to two-dimensional COFs but can also be extended to three-dimensional COFs, offering a approach for fabricating luminescent three-dimensional crystalline COF films. Moreover, the incorporation of amphiphilic chains rendered the resulting COF films with high processability and flexibility. For example, amphiphilic COF films could be bent or twisted repeatedly without damage (Supplementary Fig. S35). In contrast, the non-substituted **Ph-TAPB-COF** (lacking amphiphilic chains) produced through the same process resulted in brittle film formation (Supplementary Fig. S24), indicative of weak mechanical properties. Intriguingly, the free-standing film of **PhPeMa-TOB-COF** not only exhibited remarkable flexibility but also demonstrated stretchability. The maximum stress and strain at the membrane break

of **PhPeMa-TOB-COF** reached 10.8 MPa and 8.9%, respectively (Supplementary Fig. S36). These findings underscore the pivotal role of amphiphilic chains in enhancing the mechanical properties of free-standing COF films.

To elucidate the proposed mechanism for the formation of COF films utilizing the IPOG strategy (Fig. 4a and Supplementary Fig. S37a), time-dependent studies involving SEM, XRD, and FT-IR analyses were conducted to monitor the conversion into COF films. The interfacial preassembly process is influenced by oil-water interfacial tension and the hydrophilic and hydrophobic interactions. The results from time-dependent SEM studies demonstrate the controlled interface preassembly, oriented growth, and subsequent crystallization of the COFs. As the reaction time increased, the amphiphilic **PhPeMa-TAPB-COF** was progressively transformed from nanosheets into a uniform film (Fig. 4b). In contrast, the hydrophobic **PhPeMa-TAPB-COF** underwent preassembly into nanospheres, which subsequently transitioned into relatively rough thin films (Supplementary Fig. S37b). Notably, crystallinity was observed to increase over time, which can be attributed to enhanced π-π stacking during the crystal growth process (Fig. 4c). Further confirmation of the compositions was achieved through Fourier transform infrared (FT-IR) spectroscopy (Fig. 4d). It was noted that as the reaction progressed, the characteristic peaks of the precursor aldehyde and amine began to diminish, while new peaks corresponding to the formation of new bonds emerged. The disappearance of the N-H and aldehyde -C=O stretching bands at 3342 cm$^{-1}$ and 1682 cm$^{-1}$, respectively, indicates the absence of the starting precursors in the resulting films. The characteristic stretching frequencies for the COF film at 1609 cm$^{-1}$ (-C=N) and 1398 cm$^{-1}$ (N-Ar)

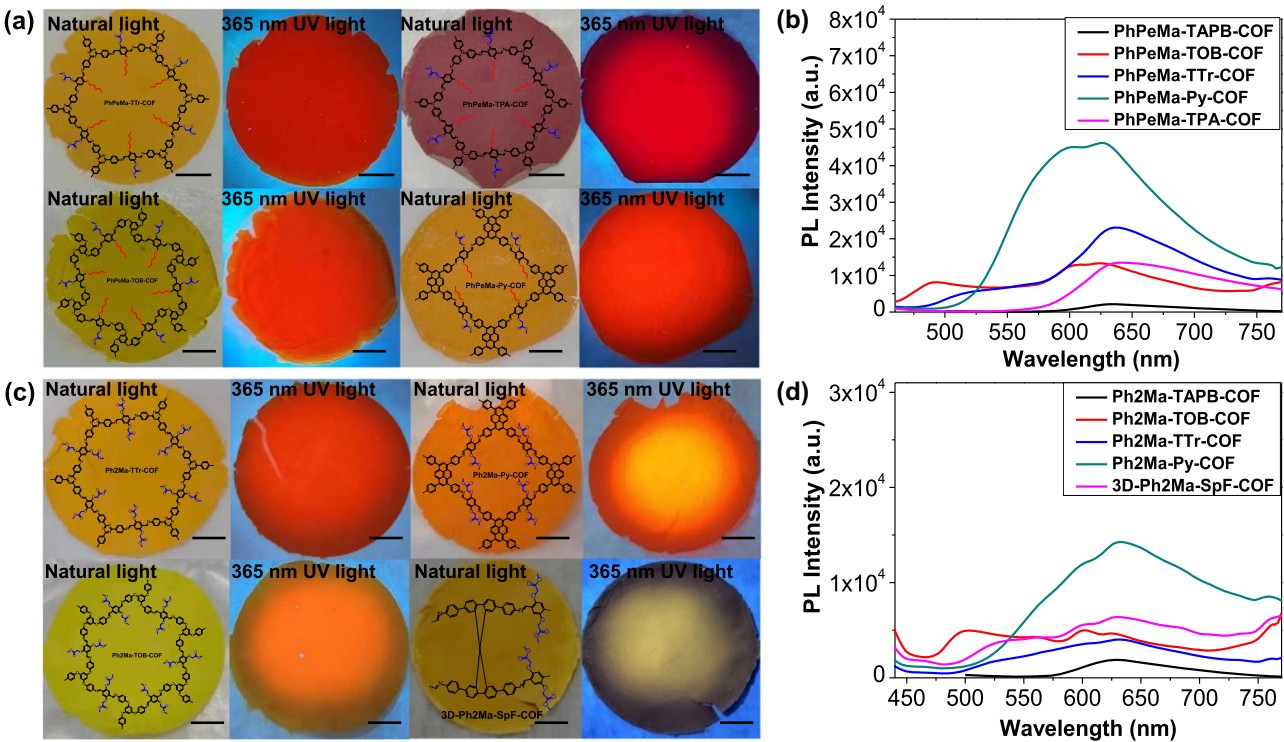

**Fig. 3 | Characterizations of COF films.** A Photographs (**a**) and PL spectra (**b**) of amphiphilic luminescent COF (**PhPeMa-TTr-COF**, **PhPeMa-TPA-COF**, **PhPeMa-TOB-COF**, and **PhPeMa-Py-COF**) films. Photographs (**c**) and PL spectra (**d**) of hydrophilic luminescent COF (**Ph2Ma-TTr-COF**, **Ph2Ma-Py-COF**, **Ph2Ma-TOB-COF**, and **3D-Ph2Ma-SpF-COF**) films.

exhibited a gradual increase corresponding to the extended reaction time.

To explore the potential for electroluminescent (EL) device applications, solution-processed COF-based organic light-emitting diodes (OLEDs) were fabricated with the configuration of ITO/PEDOT:PSS (50 nm)/PTAA:COF (100 nm)/TPBi (45 nm)/LiF (1 nm)/Al (100 nm) (Fig. 5a). As shown in Fig. 5b, all the COF-based OLEDs based on **PhPeMa-TAPB-COF**, **Ph2Ma-TAPB-COF**, and **PhPeMa-Py-COF** as emitting layers exhibited remarkably low turn-on voltages ($V_{on}$ <3 V). The maximum luminance ($L_{max}$) values of the device based on **PhPeMa-TAPB-COF, Ph2Ma-TAPB-COF**, and **PhPeMa-Py-COF** are 1424 cd m$^{-2}$, 1031 cd m$^{-2}$, and 532 cd m$^{-2}$, respectively, surpassing the luminance levels of conventional commercial display devices (100 cd m$^{-2}$)[22]. Different batches of devices were prepared based on the three kinds of COFs, and the luminance of the devices remained basically the same in different manufacturing processes (Supplementary Fig. S38), indicating that the COF-based OLEDs have good repeatability. Under the same driving voltage, the current density of the device based on **PhPeMa-TAPB-COF** is observed to be 5‰ of that of the recently reported **IISERP-COF7**-based devices[23] and 2‰ of the strontium metal organic framework (MOF)-based devices[24], indicating that amphiphilic COFs show more efficient current injection. To the best of our knowledge, this represents the highest performance achieved to date in both MOF-based and COF-based devices (Supplementary Table S2)[25–27]. The current efficiency of amphiphilic **PhPeMa-TAPB-COF** devices was measured at 0.91 cd A$^{-1}$, nearly double that of hydrophilic **Ph2Ma-TAPB-COF** devices (Fig. 5c), indicating the significant influence of hydrophilic and hydrophobic chains on the optoelectronic performance of COFs.

Notably, the EL spectra of COF-based OLEDs exhibited significant variation with increasing voltage, transitioning from deep blue to blue and finally to white emission. As the voltage increased, the short-wavelength peak (~415 nm) gradually diminished, while the long-wavelength peak (~650 nm) exhibited a corresponding increase

(Fig. 5d). A comparison was conducted between the electroluminescence spectra of **PhPeMa-TAPB-COF**-based OLEDs and the photoluminescence spectra of **PTAA** and **PhPeMa-TAPB-COF** (Supplementary Fig. S39). It was observed that the short-wavelength emission aligns with the emission peak of **PTAA**, whereas the long-wavelength emission corresponds to the emission peak of **PhPeMa-TAPB-COF**. At high voltages, a greater number of charge carriers are delivered to the **PhPeMa-TAPB-COF**, resulting in enhanced emission. Additionally, OLEDs based on **PhPeMa-TAPB-COF** and **PhPeMa-Py-COF** also displayed exceptional voltage-dependent multicolor emission (Supplementary Fig. S40 and Fig. S41). The Commission Internationale de l'Eclairage (CIE) coordinates indicated dominant blue emission from 4 V to 8 V, transitioning to white emission with further voltage increments (Fig. 5e, Supplementary Fig. S42 and Fig. S43). The CIE coordinates (Supplementary Table S2) for OLEDs using **PhPeMa-TAPB-COF, Ph2Ma-TAPB-COF**, and **PhPeMa-Py-COF** were determined to be (0.33, 0.39), (0.34, 0.40), and (0.32, 0.39), respectively, closely aligning with the standard white light coordinates (0.33, 0.33). The voltage-dependent electroluminescent devices offer a straightforward method for producing color-tunable displays. Furthermore, the correlated color temperature (CCT) of these COF-based OLEDs could be adjusted between 4500 K and 10,000 K with varying voltages (Fig. 5f). While possessing the same structure as **PhPeMa-TAPB-COF** synthesized via IPOG, OLEDs fabricated using sol-**PhPeMa-TAPB-COF** synthesized through the solvothermal method were found to be ineffective in operation. SEM images revealed that **PTAA** and **PhPeMa-TAPB-COF** blending films displayed lower roughness compared to sol-**PhPeMa-TAPB-COF** blending films (Supplementary Fig. S44). The presence of large nano-bulks in sol-**PhPeMa-TAPB-COF** films impacted the formation of the interface and electrode layers in the device, leading to operational failures. This highlights the advantageous role of the IPOG strategy in facilitating the formation of a two-dimensional ordered nanosheet structure in COFs for OLED preparation. The microstructure of the blended films was further characterized by AFM,

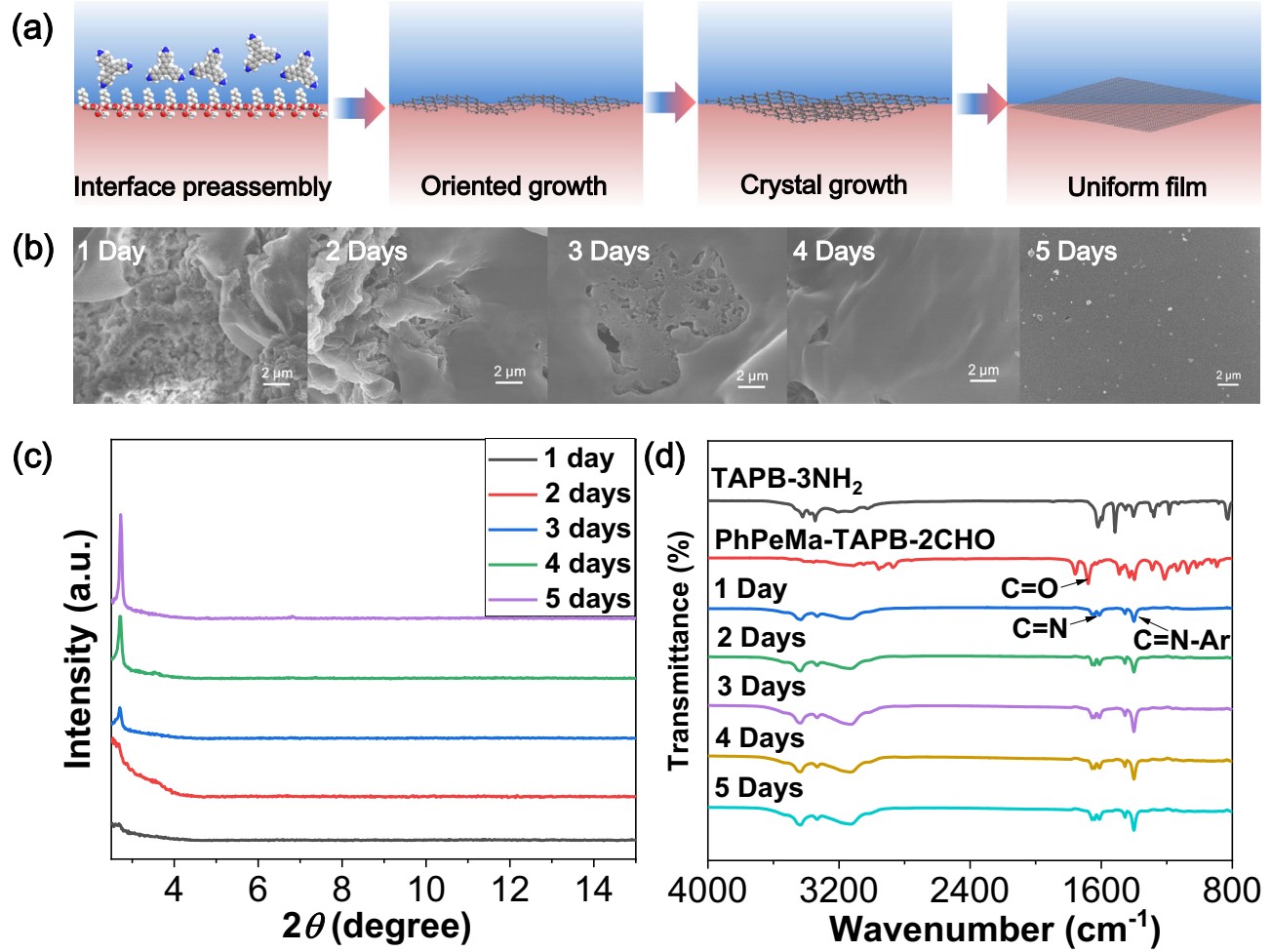

**Fig. 4 | Proposed mechanism of COF films. a** Proposed mechanism for **PhPeMa-TAPB-COF** film formation using IPOG strategy. Time-dependent SEM (**b**), XRD (**c**), and FT-IR (**d**) studies provide support for the proposed mechanism.

and the COF films synthesized by solvothermal method showed higher roughness (Supplementary Fig. S45). Additionally, COFs synthesized using the IPOG strategy demonstrate promising potential for application in flexible OLED devices due to their structured two-dimensional nanosheets. The flexible COF-based OLEDs were successfully fabricated, showcasing stable electroluminescence performance even under external bending, internal bending, and folding conditions (Supplementary Fig. S46). These flexible OLEDs demonstrated low turn-on voltages (Supplementary Fig. S47) and appropriate current efficiency (Supplementary Fig. S48). Although the full potential of these COFs in flexible OLEDs has not yet been fully explored to optimize performance due to electrode and interface layer mismatches, the results present COFs as a highly promising material for constructing flexible OLEDs, opening up an avenue for future flexible display and lighting applications.

In conclusion, a comprehensive, straightforward, and effective strategy has been devised for fabricating diverse flexible crystalline COF films using the IPOG method for electroluminescent devices. By addressing concerns surrounding thin film production, luminescence, and electronic properties, the potential application of COFs as luminescent materials has been broadened. The maximum luminance of COF-based OLEDs reaches 1424 cd m$^{-2}$, representing unprecedented performance achieved by COF-based and MOF-based devices. The development of the flexible OLEDs based on COFs unlocks the application potential of COFs in flexible electronics. Through meticulous adjustment of hydrophilic/hydrophobic chains, conjugated backbones, topological geometries, and linked bonds, it is anticipated that the electroluminescent properties from COFs can be further enhanced to pave the way for high-performance devices. Of particular note, voltage-dependent multicolor OLEDs have been crafted using these luminescent COFs, providing a feasible scheme for the preparation of tunable multicolor display devices. Our study shed light on the design and development of next-generation flexible crystalline materials to combine excellent mechanical flexibility with high optoelectronic performance for flexible electronics.

## Methods

### Interface preassembly oriented growth of Ph2Pe-TAPB-COF films

In a beaker containing 10 mL of aqueous acetic acid (6 M), slowly added 5 mL of **Ph2Pe-2CHO** (13.8 mg, 0.045 mmol) solution (1,4-dioxane/1,3,5-trimethylbenzene v:v = 3:2). Standing for 5 min, **Ph2Pe-2CHO** is preassembled into a uniform film at the oil-water interface under the action of hydrophobic and hydrophilic chains. The addition of solution monomer should be slow, which is conducive to the formation of oil-water interface. Then slowly add 5 mL of **TAPB-3NH₂** (10.5 mg, 0.03 mmol) solution (1,4-dioxane/1,3,5-trimethylbenzene v:v = 3:2). After the reaction at room temperature for 5 days, the film at the interface was removed and soaked in tetrahydrofuran and acetone for 2 days. The yellow film (**Ph2Pe-TAPB-COF**) was obtained by vacuum drying (yield 76%). According to the above-described interface preassembly oriented growth process, non-substituted **Ph-TAPB-COF** films were prepared.

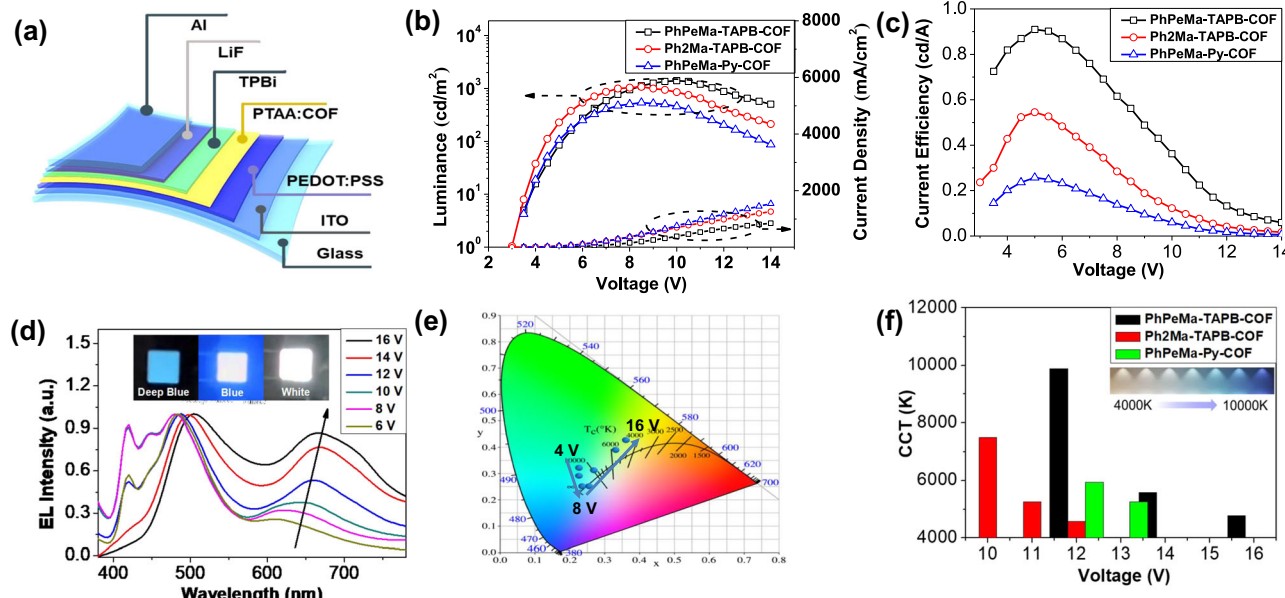

**Fig. 5 | Fabrication and characterizations of COF-based OLEDs. a** The configuration of COF-based OLEDs. **b** Current density-luminance-voltage (*J-L-V*) characteristics of COF-based OLEDs. **c** Current efficiency-voltage curves of COF-based OLEDs. EL spectra (**d**) and CIE diagram (**e**) of **PhPeMa-TAPB-COF**-based OLEDs with different driving voltages. **f** CCT of COF-based OLEDs with different driving voltage.

## Interface preassembly oriented growth of PhPeMa-TAPB-COF films

In a beaker containing 10 mL of aqueous acetic acid (6 M), slowly added 5 mL of **PhPeMa-2CHO** (13.8 mg, 0.045 mmol) solution (1,4-dioxane/1,3,5-trimethylbenzene v:v = 3:2). Standing for 5 min, **PhPeMa-2CHO** is pre-assembled into a uniform film at the oil-water interface under the action of hydrophobic and hydrophilic chains. The addition of solution monomer should be slow, which is conducive to the formation of oil-water interface. Then slowly add 5 mL of **TAPB-3NH$_2$** (10.5 mg, 0.03 mmol) solution (1,4-dioxane/1,3,5-trimethylbenzene v:v = 3:2). After the reaction at room temperature for 5 days, the film at the interface was removed and soaked in tetrahydrofuran and acetone for 2 days. The yellow film (**PhPeMa-TAPB-COF**) was obtained by vacuum drying (yield 83%). According to the above-described interface preassembly oriented growth process, amphiphilic luminescent COF films **PhPeMa-TTr-COF, PhPeMa-TPA-COF, PhPeMa-TOB-COF**, and **PhPeMa-Py-COF** were prepared.

**Solvothermal synthesis of PhPeMa-TAPB-COF powders:** **PhPeMa-2CHO** (13.8 mg, 0.045 mmol) and **TAPB-3NH$_2$** (10.5 mg, 0.03 mmol) were weighed into a Pyrex tube, and the mixture was added into 6 mL of 1,4-dioxane and 4 mL of 1,3,5-trimethylbenzene. The mixture was sonicated for 5 min to form a uniform solution. Then, 2 mL of aqueous acetic acid (6 M) were added to the reaction solution. The Pyrex tube was flash frozen in a liquid nitrogen bath, evacuated to an internal pressure of ca. 19.0 mbar. The reaction mixture was heated at 120 °C for 120 h. The obtained precipitate was isolated by filtration and washed with acetone and *n*-hexane for 4 h, respectively. The obtained powder was immersed in anhydrous acetone, and the solvent was exchanged with fresh acetone several times. The obtained sample was then transferred to vacuum chamber at 75 °C for 24 h, yielding yellow powders (yield 56%).

## Interface preassembly oriented growth of Ph2Ma-TAPB-COF films

In a beaker containing 10 mL of aqueous acetic acid (6 M), slowly added 5 mL of **Ph2Ma-2CHO** (13.9 mg, 0.045 mmol) solution (1,4-dioxane/1,3,5-trimethylbenzene v:v = 3:2). Standing for 5 min, **Ph2Ma-2CHO** is

pre-assembled into a uniform film at the oil-water interface under the action of hydrophobic and hydrophilic chains. The addition of solution monomer should be slow, which is conducive to the formation of oil-water interface. Then slowly add 5 mL of **TAPB-3NH$_2$** (10.5 mg, 0.03 mmol) solution (1,4-dioxane/1,3,5-trimethylbenzene v:v = 3:2). After the reaction at room temperature for 5 days, the film at the interface was removed and soaked in tetrahydrofuran and acetone for 2 days. The red film (**Ph2Ma-TAPB-COF**) was obtained by vacuum drying (yield 86%). According to the above-described interface preassembly oriented growth process, bihydrophilic luminescent COF films **Ph2Ma-TTr-COF, Ph2Ma-Py-COF, Ph2Ma-TOB-COF**, and **3D-Ph2Ma-SpF-COF** were prepared.

## Data availability

All data needed to evaluate the conclusions given in the paper are present in the Article and Supplementary Information. Any additional data related to this paper may be requested from the corresponding authors.

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

## Acknowledgements

We acknowledge financial support from the National Key Research and Development Program of China (2023YFB3608900, 2024YFB3612500, 2024YFB3612600), Basic Research Program of Jiangsu Province (BK20243057), the National Natural Science Foundation of China (21835003, 62005126, 61704077), the Natural Science Foundation of Jiangsu Province (BE2019120), the China Postdoctoral Science Foundation (2019M650121), the Postdoctoral Science Foundation of Jiangsu Province (1701135B), Program for Jiangsu Specially-Appointed Professor (RK030STP15001), the NUPT "1311 Project" and the NUPT Scientific Foundation (NY220152, NY219021), State Key Laboratory of Organic Electronics and Information Displays (GZR2024010001), the Leading Talent of Technological Innovation of National Ten-Thousands Talents Program of China.

## Author contributions

X.-C.L. and W.-Y.L. conceived the idea and designed the experiments. X.C.L., H.S., Z.W. and W.Y. contributed to the material synthesis and the data analysis. X.C.L., Q.W. and C.W. performed NMR, XRD, and BET measurements. J.C. and Q.J. performed TEM and AFM measurements. Z.W. and Q.X. performed OLED measurements. L.-J.H. contributed to the mechanism analysis. X.C.L., H.S., W.H. and W.-Y.L. wrote the manuscript. All authors discussed the results and commented on the manuscript. W.-Y.L. led the project.

## Competing interests

The authors declare no competing interests.
