## [Transparent Peer Review file · Nature Communications]

Interface Preassembly Oriented Growth Strategy Towards Flexible Crystalline Covalent Organic Framework Films for OLEDs

Corresponding Author: Professor Wen-Yong Lai

Version 0:

Reviewer comments:

Reviewer #1

(Remarks to the Author)

This manuscript introduces a method for the interface preassembly-oriented growth (IPOG) of flexible crystalline covalent organic framework (COF) thin films. The growth process is optimized through strategic control of hydrophilic and hydrophobic interactions, which enhances the luminescence of the films. Additionally, the potential of these luminescent COF films for high-performance organic light-emitting diodes (OLEDs) is evaluated. I think this paper presents an interesting strategy for fabricating COF films with good performance. However, there are still some problems needs to be further addressed. The paper can be published in Nature Communications after major revision.

1. There are notable differences in the preparation methods for COF films compared to COF powders. The current PXRD tests are conducted on powder samples; therefore, it is recommended to perform PXRD analysis directly on the COF films to accurately assess their crystallinity. Additionally, it would be pertinent to investigate whether the observed poor crystallinity of Ph2Pe-TAPB-COF is associated with the presence of hydrophobic groups. It is also important to explore whether the crystallinity of the COF films has an impact on their luminescence performance.
2. The structure of COFs is not clearly articulated, and there is a lack of a defined structure-activity relationship. Please clarify why the main peak positions in the experimental PXRD patterns of Ph2Pe-TAPB-COF and Ph2Ma-TAPB-COF align with the models for both AA and AB stacking modes.
3. It is recommended to supplement the AFM data to further illustrate the surface properties of COF films.
4. The paper does not demonstrate that the COF thin films have controllable thickness, as mentioned in the introduction. While different films exhibit varying thicknesses, the same film does not control its thickness under different conditions. Additionally, it is important to investigate whether the thickness of the films influences the performance of COF-based OLEDs.

Reviewer #2

(Remarks to the Author)

The authors reported an innovative method for fabricating flexible crystalline covalent organic framework films through an interface preassembly oriented growth strategy. The research successfully balances the conflicting properties of flexibility and crystallinity, achieving a breakthrough in the development of electroluminescent materials. The integration of hydrophilic and hydrophobic interactions to control COF film growth is a novel concept that enhances both the mechanical flexibility and optoelectronic performance of COFs. The application of these COF films in organic light-emitting diodes demonstrates their potential for next-generation flexible electronics. Overall, this study provides valuable insights into the design and application of COF films, making it a candidate for publication in "Nature Communications".

1. Please provide a more detailed comparison between the IPOG strategy and existing liquid-liquid interface synthesis methods.
2. The font size in Fig. 1b is currently too small to ensure readability. Please increase the font size to improve clarity and make it accessible for readers.
3. The author tested the water contact angles of the COF films. What effect does the introduction of hydrophilic chains have on the size of the contact angles between the two sides of the films? A discussion on this effect would provide deeper

insights into the surface properties.

4. Please update the literatures to incorporate recent studies on COF films.

5. Please check and improve the style according to the requirements of Nature Communications. A thorough review of formatting, citations, and overall presentation will be beneficial.

Reviewer #3

(Remarks to the Author)

In the present study, authors have demonstrated an interface preassembly-oriented growth (IPOG) strategy by introducing hydrophilic and hydrophobic side groups into the aldehyde linker to fabricate a series of covalent organic framework (COF) films. The COF films with amphiphilic and hydrophilic linkers show good mechanical stability and high efficiency in electroluminescence. However, there are several issues related to the material's characterization and its optoelectronic properties. Hence, I am unable to endorse the present manuscript for publication. A thoroughly revised manuscript with new data and analyses could be resubmitted to Nature Communications for further review.

1. A key aspect of this study is the use of hydrophobic-hydrophilic side-chain-functionalized dialdehyde linkers. It is essential that these linkers are thoroughly characterized and the synthetic and purification protocols are described in detail. Please provide the ¹³C NMR and mass spectrometric data (HRMS) for Ph₂Pe-2CHO, PhPeMa-2CHO, and Ph₂Ma-2CHO linkers. The reaction conditions, including solvents, reaction time, and acid concentration, should be clearly indicated in the scheme (Fig. 1a). The blue droplet should be labeled as aldehyde linkers, while the red droplet represents amine linkers in a mixture of organic solvents.

2. Synthesis and experimental conditions must be provided in such a detailed way so that everything can be reproduced by other interested readers/ researchers. Currently, this is not the case.

3. Why was 1,4-dioxane selected as one of the organic solvents (1,4-dioxane: mesitylene = 3:2 v/v) in the interfacial polymerization? Its miscibility with water could disrupt the sharp interface between the aqueous and organic phases.

4. "All amphiphilic and hydrophilic COF films exhibited high crystallinity, with refined PXRD patterns consistent with experimental profiles (Fig. S8-S15)." Most of the COF films in this study exhibit poor PXRD patterns, with high *R_p* and *R_w* values exceeding 10% (Fig. S10-S15). Therefore, the claim of high crystallinity for these COF films is not well-supported. Please calculate the crystalline domain size for the COF films by Scherrer analysis. Please check the following literature: J. Am. Chem. Soc. 2017, 139, 37, 12911-12914; J. Am. Chem. Soc. 2020, 142, 35, 14957-14965; Chem. Commun. 2023, 59, 13639-13642; Angew. Chem. Int. Ed., 2023, 62, e202219083. Some of these papers, which are relevant to interfacial COF fabrication, could be cited.

5. The ordered structure of the COF films with eclipsed (AA) stacking can be further validated through pore size distribution analysis analyzing the BET adsorption isotherms. Please provide the surface area and porosity data to better understand the ordered structure of the COF films.

6. Why do COFs with hydrophilic and amphiphilic linkers exhibit a more "ordered" structure compared to COF films with hydrophobic side chains? No experimental analysis or explanation was provided for this observation. To justify the "interface preassembly oriented growth" mechanism, a time-dependent COF growth study using spectroscopic and microscopic investigations should be conducted. This point must be addressed critically. Please check the following literature: Langmuir, 2024, 40, 16419-16429; Angew. Chem. Int. Ed., 2023, 62, e202219083; J. Membr. Sci. 2022, 650, 120431; J. Am. Chem. Soc., 2019, 141, 20371; Nat. Chem. 2019, 11, 994-1000.

7. "To explore the potential for electroluminescent (EL) device applications, solution-processed COF-based organic light-emitting diodes (OLEDs) were fabricated with the configuration of ITO/PEDOT:PSS (50 nm)/PTAA:COF (100 nm)/TPBi (45 nm)/LiF (1 nm)/Al (100 nm) (Fig. 4a)." What does the value in the parentheses [e.g., (100 nm)] indicate? Does it refer to the thickness of the layers? How was the thickness of each layer controlled?

8. "With the increase of voltage, the short-wavelength peak (~415 nm) of PhPeMa-TAPB-COF-based devices gradually diminished, while the long-wavelength peak (~650 nm) increased gradually (Fig. 4d)." The explanation for this observation is missing.

9. "The maximum luminance (*L_{max}*) values of the device based on PhPeMa-TAPB-COF, Ph₂Ma-TAPBCOF, and PhPeMa-Py-COF are 1424 cd m⁻², 1031 cd m⁻², and 532 cd m⁻², respectively, surpassing the luminance levels of conventional commercial display devices (100 cd m⁻²)." Why does PhPeMa-Py-COF exhibit a comparatively low luminance value despite having a higher photoluminescence intensity (Fig. 4b)?

10. Luminance values should be presented with error bars by triplicate measurements using samples obtained from three different batches of fabrication.

11. "Furthermore, the presence of hydrophilic and hydrophobic chains does not disrupt the interlayer conjugation of COFs, aiding in enhancing charge carrier transport." What is meant by "interlayer conjugation"?

12. "While possessing the same structure as PhPeMa-TAPB-COF synthesized via IPOG, OLEDs fabricated using sol-PhPeMa-TAPB-COF synthesized through the solvothermal method were found to be ineffective in operation. SEM images revealed that PTAA and PhPeMa-TAPB-COF blending films displayed lower roughness compared to sol-PhPeMa-TAPBCOF blending films (Fig. S28)." Fig. S28d and S28e are unclear. Please provide AFM images to compare the surface roughness of the PTAA-blended PhPeMa-TAPB-COF film synthesized via the IPOG method with that of the PTAA-blended PhPeMa-TAPB-COF synthesized via the solvothermal route.

13. Minor correction: There are several typos in the manuscript. For example, "Under the same driving voltage, the current density of PhPeMa-TAPB-COF-based device is 5‰ of that of the recently reported IISERP-COF7-based and 2‰ of the strontium metal-organic framework (MOF)-based devices, indicating that amphiphilic COFs show more efficient current injection."

Reviewer #4

(Remarks to the Author)

Version 1:

Reviewer comments:

Reviewer #1

(Remarks to the Author)

I am OK with the revision and recommend the publication.

Reviewer #2

(Remarks to the Author)

Accept.

Reviewer #3

(Remarks to the Author)

The authors have addressed most of the reviewers' queries, conducted several new experiments to validate their claims, and revised the manuscript and SI accordingly. The clarity and quality of the manuscript has been improved significantly. However, the newly conducted experiments and added text have raised additional questions that need to be addressed before the manuscript can be considered for publication in Nature Communications.

1. In response to Comment 4# from Reviewer 1 and Comment 7# from Reviewer 3, it remains unclear how the thickness of the COF films was precisely controlled. The authors stated that the thickness was regulated by the initial concentration of the monomers. Please specify the starting concentrations of the monomers used to achieve COF films with thicknesses of 100, 200, and 300 nm.
2. As indicated above, please provide specific details in the caption of Fig. S24 (monomer concentrations, reaction times).
3. In response to Comment 7# from Reviewer 3, the authors stated: "(a) For the layers formed from solutions of COFs and other materials (e.g., PEDOT:PSS, PTAA:COF), spin coating was utilized. This technique allows for precise control over the thickness of the deposited layers through the adjustment of parameters such as spin speed, solution concentration, and viscosity." However, the term "solution of COFs" is unclear. How can a pre-synthesized COF film (insoluble) be used for spin coating? This statement lacks clarity. If the COF synthesis methodology used to fabricate the OLED device differs from the interfacial technique, it raises concerns about the consistency of physical properties (e.g., crystallinity, porosity, surface morphology) of the COF film in the device. Consequently, the structure-property relationship could be misleading. Please address these concerns critically.
4. In response to Comment 5# from Reviewer 3, the specific BET surface area of all the COFs is reported to be very low (13-45 m²/g, significantly less surface area compared to porous COF films obtained through standard interfacial polycondensation). The reason for such low values is unclear. Please clarify.
5. Please specify in the manuscript which model was used to determine the pore size distribution of the COFs. Additionally, the theoretical pore size of these COFs should be less than 3.3 nm. Please provide an explanation or comments on this discrepancy.
6. Please avoid representing the BET-specific surface area up to a decimal place. Even though, based on the fitting of the BET equation, we obtain BET surface area up to multiple decimal places, the sensitivity of the instrument is unlikely to differentiate 0.1 m²/g.
7. There are still some errors in the manuscript. For instance, References 14 and 21 are identical.

Reviewer #4

(Remarks to the Author)

Version 2:

Reviewer comments:

Reviewer #3

(Remarks to the Author)

The authors have addressed all the concerns very carefully and critically. The clarity and quality of the manuscript are significant enough to be considered for publication in Nature Communication. However, I request that Figure R1 and Figure

R2 (as a normalized PXRD plot of COF dispersion/ nanocrystalline powder obtained from a thin film and pristine thin film) be included in the supporting information for better clarity and understanding.
This manuscript may not require further review by this reviewer.

Reviewer #4

(Remarks to the Author)

Point-by-point response to the reviewers' comments

Reviewer #1:

Comment: *"This manuscript introduces a method for the interface preassembly-oriented growth (IPOG) of flexible crystalline covalent organic framework (COF) thin films. The growth process is optimized through strategic control of hydrophilic and hydrophobic interactions, which enhances the luminescence of the films. Additionally, the potential of these luminescent COF films for high-performance organic light-emitting diodes (OLEDs) is evaluated. I think this paper presents an interesting strategy for fabricating COF films with good performance. However, there are still some problems needs to be further addressed. The paper can be published in Nature Communications after major revision."*

Author Reply: We greatly appreciate the reviewer's highly positive and constructive comments for this work. We have considered the valuable remarks seriously and revised carefully our manuscript accordingly. The answers to the comments are as follows.

Comment #1: *"There are notable differences in the preparation methods for COF films compared to COF powders. The current PXRD tests are conducted on powder samples; therefore, it is recommended to perform PXRD analysis directly on the COF films to accurately assess their crystallinity. Additionally, it would be pertinent to investigate whether the observed poor crystallinity of Ph₂Pe-TAPB-COF is associated with the presence of hydrophobic groups. It is also important to explore whether the crystallinity of the COF films has an impact on their luminescence performance."*

Author Reply: Thank you for your insightful comments and suggestions. We acknowledge the importance of performing XRD analysis directly on the COF films to accurately assess their crystallinity. The manuscript details the use of XRD to confirm the crystalline nature of the COF films, which exhibited intense diffraction peaks indicative of crystallinity. In this manuscript, the COF films prepared using the IPOG method underwent film XRD analysis rather than powder XRD. The previous mention of PXRD (powder X-ray

diffraction) in this manuscript may lead to potential misunderstandings for readers. Therefore, this reference has been removed in the revised manuscript.

To investigate the relationship between the observed poor crystallinity of **Ph2Pe-TAPB-COF** and the presence of hydrophobic groups, a comparative analysis was conducted on COFs with varying hydrophobic properties to understand their impact on crystallinity and performance. The manuscript discusses the synthesis of hydrophobic **Ph2Pe-TAPB-COF**, amphiphilic **PhPeMa-TAPB-COF**, and hydrophilic **Ph2Ma-TAPB-COF** to explore the impact of hydrophilic/hydrophobic chains on film formation. It was observed that **Ph2Pe-TAPB-COF** had a less favorable pawley-refined XRD pattern with residuals, suggesting a potential influence of hydrophobic groups on crystallinity. The existence of hydrophobic chains reduces the solubility in aqueous solution, making it difficult to form an effective preassembly structure at the interface, resulting in reduced crystallinity.

We have also explored the impact of crystallinity on the luminescence performance of the COF films. Our results indicate a correlation between the crystallinity of the COF films and their luminescence properties. As demonstrated in Fig. 2d, the amphiphilic **PhPeMa-TAPB-COF**, which exhibits high crystallinity, also shows stronger fluorescence emission compared to the hydrophobic **Ph2Pe-TAPB-COF** and hydrophilic **Ph2Ma-TAPB-COF**. The photoluminescence quantum yield of **PhPeMa-TAPB-COF** was found to be 60 times greater than that of **Ph2Pe-TAPB-COF** in solid film, highlighting the importance of crystallinity for enhanced luminescence. However, crystallinity is not the most important factor affecting the luminescence properties of COF. As shown in Fig. 2a, inhibition of intramolecular rotation and interlayer π - π interaction, and reduction of non-radiative transition caused by aggregation are the main factors affecting the luminescence performance of COF.

Comment #2: *"The structure of COFs is not clearly articulated, and there is a lack of a defined structure-activity relationship. Please clarify why the main peak positions in the*

experimental PXRD patterns of Ph2Pe-TAPB-COF and Ph2Ma-TAPB-COF align with the models for both AA and AB stacking modes."

Author Reply: Thank you for your valuable comments. The structure-activity relationship in the manuscript is primarily governed by the hydrophilic and hydrophobic interactions that influence the crystallization and luminescent properties of the COF films. The introduction of hydrophilic and hydrophobic chains into the monomers affects the self-assembly at the liquid-liquid interface, which in turn influences the orientation and alignment of the building blocks, leading to variations in film formation and luminescence efficiency. To further elucidate the structure-activity relationship, we have systematically investigated the impact of hydrophilic and hydrophobic chains on the luminescence properties of the COF films. As demonstrated in Fig. 2d, the amphiphilic **PhPeMa-TAPB-COF**, which has a more pronounced peak alignment with both AA stacking modes, exhibits a significantly higher photoluminescence quantum yield compared to the hydrophobic **Ph2Pe-TAPB-COF**. This observation underscores the influence of the structure on the luminescent properties of the COFs.

According to the definition of the International Union of Pure and Applied Chemistry (IUPAC), mesoporous materials are porous materials with pore sizes in the range of 2-50 nm. According to Bragg's equation $d_{hkl} = \lambda / (2 \sin \theta_{hkl})$, the corresponding $2\theta = 0^\circ \sim 4^\circ$. By XRD, the main peak position of the result COF films is about 2.7 degrees, which can be determined that these COFs are mesoporous material. Therefore, the small-angle XRD peaks of COFs materials mainly provide relevant information about the pore size and pore structure of mesoporous materials. Large angle XRD peaks can reveal the crystal structure characteristics of COF materials, including crystal surface spacing, packing structure, crystal symmetry, etc. The position and intensity of these large angle peaks are directly related to the molecular arrangement. The positions of the main peaks in the XRD patterns of **Ph2Pe-TAPB-COF** and **Ph2Ma-TAPB-COF**, indicating that these COFs are mesoporous materials. The simulated XRD is obviously different at large angular peaks.

Comment #3: "It is recommended to supplement the AFM data to further illustrate the surface properties of COF films."

Author Reply: Thank you once again for the valuable suggestion to supplement the AFM data to further illustrate the surface properties of COF films. In response to this recommendation, AFM measurements have been conducted on the COF films, yielding detailed information regarding surface roughness and morphology. The AFM images of the **PhPeMa-TAPB-COF** films reveal a uniform surface morphology, which aligns with the observations from SEM. The root mean square (RMS) roughness values obtained from the AFM measurements are relatively low, indicating the high quality of the COF film surface. Furthermore, the AFM data demonstrate that the introduction of hydrophilic and hydrophobic chains significantly influences the surface properties of the COF films. The amphiphilic COF films, such as **PhPeMa-TAPB-COF**, exhibit a more uniform surface in comparison to **Ph2Pe-TAPB-COF** and **Ph2Ma-TAPB-COF** films, which is consistent with the SEM observations that highlight the enhanced film-forming properties of the amphiphilic COFs. We have included the AFM images and the corresponding roughness data in the revised manuscript. The changes are as follows: "The surface properties of COF films were further characterized using atomic force microscopy (AFM) analysis (Supplementary Fig. S26). The results indicate that the amphiphilic **PhPeMa-TAPB-COF** films exhibit reduced roughness compared to the **Ph2Pe-TAPB-COF** and **Ph2Ma-TAPB-COF** films. This observation suggests that the introduction of amphiphilic chains facilitates the formation of a more uniform film morphology." (Page 8)

Figure S26. AFM images of **Ph2Pe-TAPB-COF** (a), **PhPeMa-TAPB-COF** (b), and **Ph2Ma-TAPB-COF** films (c).

Comment #4: *"The paper does not demonstrate that the COF thin films have controllable thickness, as mentioned in the introduction. While different films exhibit varying thicknesses, the same film does not control its thickness under different conditions. Additionally, it is important to investigate whether the thickness of the films influences the performance of COF-based OLEDs."*

Author Reply: Thank you for your comments and for pointing out the need for clarification regarding the controllability of the COF thin film thickness and its impact on the performance of COF-based OLEDs. In this study, we have indeed shown that the thickness of the COF films can be controlled by adjusting the monomer concentrations and reaction times during the interface preassembly oriented growth (IPOG) process. This controllability is a significant aspect of our work, as it allows for the fine-tuning of film properties for specific applications. To further address your concerns, we have added additional data in the revised manuscript showing the thickness variation of the **PhPeMa-TAPB-COF** films under different synthesis conditions, such as varying monomer concentrations and reaction time (Fig. S24). Under the same reaction time, the thickness of **PhPeMa-TAPB-COF** film increases gradually with the increase of monomer concentration. When the monomer concentration is low, it is difficult to form a self-supporting film, and with the increase of monomer concentration, a transparent self-supporting film is gradually formed. When the concentration of reactive monomer was increased, the thickness of the self-supporting film continued to increase and the film became opaque. The optical microscope showed that the thickness of the film gradually increased with the increase of reaction time and the surface morphology of the film gradually became flat with the same monomer concentration. These new data clearly demonstrate our ability to control the thickness of the COF films, thus reinforcing the claim made in the introduction. We have expanded the discussion on the thickness

variation of the COF films in the revised manuscript. The changes are as follows:

“Notably, through the precise control of monomer concentrations and reaction times, the COFs can generate uniform free-standing films with controllable thickness. It was observed that the thickness of the COF films increased in response to elevated monomer concentrations and extended reaction times (Supplementary Fig. S24). Specifically, the thickness of the free-standing films can be effectively regulated by adjusting the monomer concentration. At a constant reaction time, an increase in monomer concentration resulted in a gradual transition of the film from transparent to opaque, indicating a corresponding increase in thickness.” (Page 7)

Figure S24. (a) Photographs of **PhPeMa-TAPB-COF** films after five days of reaction with different monomer concentrations. Optical micrographs (b) and SEM images (c) of **PhPeMa-TAPB-COF** films at different reaction times under the same monomer concentration.

The thickness of COF films can influence the thickness of the light-emitting layer, thereby affecting the performance of OLEDs. To investigate this relationship, additional experiments were conducted in which the thickness of the **PhPeMa-TAPB-COF** films was varied, and their impact on the thickness of the light-emitting layer of OLEDs was

measured. It was observed that as the thickness of the light-emitting layer increased, device performance also changed (Fig. R1). Specifically, the increase in the thickness of the light-emitting layer film resulted in a greater charge transport distance, generally leading to higher current density and lower overall device performance. However, it was determined that thickness is not the primary factor influencing device performance. The influence of surface roughness and synthesis method are more significant. Despite having the same structure as **PhPeMa-TAPB-COF** synthesized *via* the IPOG method, OLEDs fabricated using sol-**PhPeMa-TAPB-COF** synthesized through the solvothermal method were found to be ineffective in operation. SEM images revealed that **PTAA** and **PhPeMa-TAPB-COF** blending films displayed lower roughness compared to sol-**PhPeMa-TAPB-COF** blending films (Fig. S41). The presence of large nano-bulks in sol-**PhPeMa-TAPB-COF** films impacted the formation of the interface and electrode layers in the device, leading to operational failures.

Figure R1. Luminance-voltage characteristics of **PhPeMa-TAPB-COF**-based OLEDs with different thickness of the light-emitting layer.

Reviewer #2:

Comment: *"The authors reported an innovative method for fabricating flexible crystalline covalent organic framework films through an interface preassembly oriented growth strategy. The research successfully balances the conflicting properties of flexibility and crystallinity, achieving a breakthrough in the development of electroluminescent materials. The integration of hydrophilic and hydrophobic interactions to control COF film growth is a novel concept that enhances both the mechanical flexibility and optoelectronic performance of COFs. The application of these COF films in organic light-emitting diodes demonstrates their potential for next-generation flexible electronics. Overall, this study provides valuable insights into the design and application of COF films, making it a candidate for publication in "Nature Communications."*

Author Reply: We appreciate very much for the highly positive, insightful and constructive comments, which are helpful to improve the manuscript. We have considered very carefully the reviewer's comments and revised the manuscript accordingly. The answers to the comments are as follows.

Comment #1: *"Please provide a more detailed comparison between the IPOG strategy and existing liquid-liquid interface synthesis methods."*

Author Reply: Thank you again for your valuable comments. As shown in Fig. R2a, the IPOG strategy involves the induction of an ordered preassembly structure at the interface of organic and aqueous phases by modulating hydrophilic and hydrophobic interactions. This method allows for the controlled synthesis of free-standing flexible crystalline COF films with good luminescence properties at room temperature. The key innovation of IPOG is the synergistic modulation of hydrophilic and hydrophobic interactions, which leads to the formation of uniform and flexible crystalline COF films with controllable thickness. As shown in Fig. R2b, traditional liquid-liquid interface synthesis methods often result in COF films with irregular morphologies and limited control over thickness and crystallinity. In contrast, IPOG provides precise control over these parameters by regulating the hydrophilic and hydrophobic interactions at the interface, leading to more uniform and higher quality COF films. Moreover, IPOG-derived COF films exhibit

exceptional mechanical properties, such as flexibility and tensile strains, which are not commonly achieved with conventional synthesis methods. This is crucial for the application of COF films in flexible electronics and optoelectronic devices. The IPOG strategy has been demonstrated to be universally applicable to a variety of COF topologies and core skeletons, offering a scalable approach to the synthesis of diverse COF films with tunable properties. The IPOG strategy represents a significant advancement in the synthesis of COF films, offering improved control over film formation, enhanced luminescence properties, and superior mechanical performance compared to existing liquid-liquid interface synthesis methods. These advantages make IPOG an attractive approach for the fabrication of COF films for various applications, particularly in the field of flexible electronics.

Figure R2. The comparison between the IPOG strategy (a) and existing liquid-liquid interface synthesis methods (b).

Comment #2: "The font size in Fig. 1b is currently too small to ensure readability. Please increase the font size to improve clarity and make it accessible for readers."

Author Reply: Thank you for your suggestion to improve the readability of the font in this figure.

In response to your feedback, we have revised Fig. 1b and have increased the font size to

ensure better readability without compromising the figure's overall layout and aesthetics.

Fig. 1. (a) Synthetic scheme for the preparation of the free-standing flexible COF films. (b) The chemical structure of Ph2Pe-TAPB-COF, PhPeMa-TAPB-COF, and Ph2Ma-TAPB-COF. (c) XRD patterns of the COF films and the simulated patterns: AA stacking mode (dark cyan) and AB eclipsed stacking mode (magenta) with the experimental pattern (black) combined Pawley refined pattern (red) and the difference (blue).

Comment #3: "The author tested the water contact angles of the COF films. What effect does the introduction of hydrophilic chains have on the size of the contact angles between the two sides of the films? A discussion on this effect would provide deeper insights into the surface properties."

Author Reply: Thank you for your insightful comment and suggestion to discuss the effect of hydrophilic chains on the water contact angles of the COF films. The introduction of

hydrophilic chains into the COF structure has a significant impact on the wettability of the films, as evidenced by the changes in the contact angle measurements. The presence of hydrophilic chains in the COF films increases the hydrophilicity of the surface, which in turn affects the water contact angle. As the hydrophilic chains interact more favorably with water molecules, they reduce the contact angle by providing more polar interaction sites on the surface. We observed that with an increase in the number of hydrophilic chains, the contact angle gradually decreased. For instance, the hydrophobic **Ph2Pe-TAPB-COF** exhibited a higher contact angle compared to the amphiphilic **PhPeMa-TAPB-COF** and the hydrophilic **Ph2Ma-TAPB-COF**. Additionally, the contact angle is influenced not only by the hydrophilic and hydrophobic chains but also by the characteristics of the water-oil interface. Notably, the amphiphilic **PhPeMa-TAPB-COF** films display a pronounced difference in contact angle on either side of the water-oil interface, providing further evidence that amphiphilic monomers can facilitate the formation of interfacial preassembly structures. We have expanded the discussion on the contact angle measurements in the revised manuscript. The changes are as follows: “With increasing hydrophilic chains, the contact angle diminishes gradually, indicative of enhanced hydrophilicity (Supplementary Fig. S27). The different wetting behavior predominantly arises from the presence of hydrophilic and hydrophobic chains within the films. Amphiphilic **PhPeMa-TAPB-COF** films exhibit a significant difference in contact angle between water and oil surfaces, further demonstrating that the incorporation of hydrophilic and hydrophobic chains enhances the preassembly at the water-oil interface.” (Page 8) This additional analysis provides a more comprehensive understanding of how the surface properties of COF films can be tailored by modulating the hydrophilic and hydrophobic balance, which is a key aspect of our IPOG strategy.

Figure S27. Water contact angles (CA) between the two sides of the **Ph2Pe-TAPB-COF**, **PhPeMa-TAPB-COF**, and **Ph2Ma-TAPB-COF** films.

Comment #4: "Please update the literatures to incorporate recent studies on COF films."

Author Reply: Many thanks for this very helpful suggestion. In response to your suggestion, the manuscript has been updated to include recent studies on COF films. A thorough search of the latest literature (*Nature*, **2024**, 630, 878; *Angew. Chem. Int. Ed.*, **2024**, 63, e202409708; *J. Am. Chem. Soc.*, **2024**, 146, 14079.) was conducted, and relevant findings have been integrated into the discussion to provide a more comprehensive overview of the current state of research in this field. The citations are as follows: "The emerging field of covalent organic frameworks (COFs) has become a cornerstone in materials and chemistry, providing a versatile platform for the design of porous crystalline polymers with atomic precision¹⁻³. COFs have been extensively studied for their application in gas separation⁴, catalysis^{5,6}, energy storage^{7,8}, sensing⁹, nanogenerators,¹⁰ and photothermal therapy¹¹, with a growing emphasis on their potential application in emerging optoelectronic devices^{12,13}. Recent advances have been made in the synthesis strategies of various COFs, paving the way for the optimization of flexibility, crystallinity, and enhanced optoelectronic performance¹⁵⁻¹⁸." (Page 3)

Comment #5: "Please check and improve the style according to the requirements of *Nature Communications*. A thorough review of formatting, citations, and overall presentation will be beneficial."

Author Reply: Thank you for your constructive suggestion regarding the manuscript's style and

presentation. In response to your request, a comprehensive review of the formatting, citations, and overall presentation has been undertaken to ensure compliance with the requirements of *Nature Communications*. All specific guidelines have been meticulously followed, and adjustments have been made to enhance the clarity and professionalism of the manuscript.

Reviewer #3:

Comment: *"In the present study, authors have demonstrated an interface preassembly-oriented growth (IPOG) strategy by introducing hydrophilic and hydrophobic side groups into the aldehyde linker to fabricate a series of covalent organic framework (COF) films. The COF films with amphiphilic and hydrophilic linkers show good mechanical stability and high efficiency in electroluminescence. However, there are several issues related to the material's characterization and its optoelectronic properties. Hence, I am unable to endorse the present manuscript for publication. A thoroughly revised manuscript with new data and analyses could be resubmitted to Nature Communications for further review."*

Author Reply: We greatly appreciate the referee's highly positive and constructive comments on the work. We have considered the valuable remarks seriously and revised carefully our manuscript accordingly. These revisions have been marked in yellow in the revised manuscript. The answers for the comments are as follows.

Comment #1: *"A key aspect of this study is the use of hydrophobic-hydrophilic side-chain-functionalized dialdehyde linkers. It is essential that these linkers are thoroughly characterized and the synthetic and purification protocols are described in detail. Please provide the ¹³C NMR and mass spectrometric data (HRMS) for Ph₂Pe-2CHO, PhPeMa-2CHO, and Ph₂Ma-2CHO linkers. The reaction conditions, including solvents, reaction time, and acid concentration, should be clearly indicated in*

the scheme (Fig. 1a). The blue droplet should be labeled as aldehyde linkers, while the red droplet represents amine linkers in a mixture of organic solvents. "

Author Reply: Many thanks for this very helpful suggestion. We agree that providing detailed synthetic and purification protocols, along with the necessary ^{13}C NMR and mass spectrometric data, would be helpful for the reproducibility and credibility of our work. We have revised the synthetic and purification protocols to include detailed reaction conditions in Supplementary Information for the preparation of the **Ph2Pe-2CHO**, **PhPeMa-2CHO**, and **Ph2Ma-2CHO**. The ^{13}C NMR of **Ph2Pe-2CHO** (**Fig. S2**) shows peaks at 189.53, 155.31, 129.36, 111.70, 69.32, 28.82, 28.24, 22.46, and 14.06. ppm, which correspond to the aromatic carbons and aldehyde group. The ^{13}C NMR of **PhPeMa-2CHO** (**Fig. S5**) exhibits peaks at 189.18, 189.10, 168.58, 156.24, 153.75, 130.17, 129.06, 112.09, 112.07, 69.39, 66.02, 52.49, 28.77, 28.22, 22.44, and 14.04 ppm, reflecting the structure of the linker with both phenyl and amphipathic side chains. The ^{13}C NMR of **Ph2Ma-2CHO** (**Fig. S8**) displays peaks at 188.74, 168.43, 154.68, 129.87, 112.44, 65.92, and 52.41. ppm, indicating the presence of the dialdehyde group and hydrophilic side chains. The MS data (**Fig. S3**, **Fig. S6**, and **Fig. S9**) confirm the molecular formula with a calculated error within the acceptable range for the molecular ion peak. The reaction conditions, including solvents, reaction time, and acid have been indicated in **Fig. 1a**. We have updated **Fig. 1a** to clearly label the blue droplet as "aldehyde monomers" and the red droplet as "amine monomers", ensuring that the representation of the reactants is accurate and unambiguous.

Figure S2. ¹³C NMR of Ph2Pe-2CHO.

Figure S3. Mass spectrum of Ph2Pe-2CHO.

Figure S5. ¹³C NMR of PhPeMa-2CHO.

Figure S6. Mass spectrum of PhPeMa-2CHO.

Figure S8. ¹³C NMR of Ph2Ma-2CHO.

Figure S9. Mass spectrum of Ph2Ma-2CHO.

Fig. 1. (a) Synthetic scheme for the preparation of the free-standing flexible COF films. (b) The chemical structure of **Ph2Pe-TAPB-COF**, **PhPeMa-TAPB-COF**, and **Ph2Ma-TAPB-COF**.

Comment #2: "Synthesis and experimental conditions must be provided in such a detailed way so that everything can be reproduced by other interested readers/ researchers. Currently, this is not the case."

Author Reply: Thank you very much for the helpful suggestion. Upon your suggestion, we have revised the Supplementary Information to include comprehensive details regarding the synthesis procedures and experimental setups. We have expanded the methods section to describe each step with precision, including the reagents, concentrations, reaction conditions, and any pertinent equipment settings. We believe that these additions will meet the standards required for reproducibility and clarity in scientific communication. The revised experimental conditions are as follows:

Synthesis of **Ph2Pe-2CHO**: **Ph2OH-2CHO** (664.5 mg, 4 mmol) and anhydrous K_2CO_3 (1105.6 mg, 8 mmol) were accurately weighed and transferred into a two-mouth flask. The flask was then sealed, and nitrogen was purged to remove moisture, followed by

heating to ensure dryness. This nitrogen purge was repeated three times before adding 1-bromopentane (1449.9 mg, 9.6 mmol). Anhydrous acetonitrile (30 mL), which had been bubbled with nitrogen, was subsequently added to the flask. The reaction mixture was then heated in an oil bath at 80°C for 24 hours. After the reaction, the mixture was extracted three times. The organic layer was then concentrated under vacuum, and the residue was washed with the filter three times. The crude product was then separated and purified using a 200-300 mesh silica gel column with an elution solvent of ethyl acetate and petroleum ether in a ratio of 1:4. Finally, the product **Ph2Pe-2CHO** was obtained after vacuum drying, yielding 860.2 mg (70%). ¹H NMR (400 MHz, CDCl₃, ppm) δ 10.55 (s, 2H), 7.45 (s, 2H), 4.11 (t, J = 6.5 Hz, 4H), 1.90-1.84 (m, 4H), 1.50-1.40 (m, 8H), 0.96 (t, J = 7.1 Hz, 6H). ¹³C NMR (101 MHz, CDCl₃, ppm) δ 189.53, 155.31, 129.36, 111.70, 69.32, 28.82, 28.24, 22.46, 14.06. GC-MS (*m/z*): Calcd for C₁₈H₂₆O₄, Exact Mass: 306.18, Mol. Wt.: 306.40, Found: 306.3.

Synthesis of **PhPeMa-2CHO**: Ph2OH-2CHO (664.5 mg, 4 mmol) and anhydrous K₂CO₃ (1105.6 mg, 8 mmol) were accurately weighed and transferred into a two-mouth flask. The flask was then sealed, and nitrogen was purged to remove moisture, followed by heating to ensure dryness. This nitrogen purge was repeated three times before adding 1-bromopentane (724.9 mg, 4.8 mmol) and methyl chloroacetate (651.1 mg, 6 mmol). Anhydrous acetonitrile (30 mL), which had been bubbled with nitrogen, was subsequently added to the flask. The reaction mixture was then heated in an oil bath at 80°C for 24 hours. After the reaction, the mixture was extracted three times. The organic layer was then concentrated under vacuum, and the residue was washed with the filter three times. The crude product was then separated and purified using a 200-300 mesh silica gel column with an elution solvent of ethyl acetate and petroleum ether in a ratio of 1:3. Finally, the product **PhPeMa-2CHO** was obtained after vacuum drying, yielding 381.8 mg (31%). ¹H NMR (400 MHz, CDCl₃, ppm) δ 10.52 (s, 1H), 10.43 (s, 1H), 7.41 (s, 1H), 7.27 (s, 1H), 4.72 (s, 2H), 4.04 (t, J = 6.5 Hz, 2H), 3.74 (s, 3H), 1.80-1.76 (m, 2H), 1.35 (dd, J = 11.4, 3.4 Hz, 4H), 0.88 (d, J = 7.1 Hz, 3H). ¹³C NMR (101 MHz, CDCl₃,

ppm) δ 189.18, 189.10, 168.58, 156.24, 153.75, 130.17, 129.06, 112.09, 112.07, 69.39, 66.02, 52.49, 28.77, 28.22, 22.44, 14.04. GC-MS (m/z): Calcd for $C_{16}H_{20}O_6$, Exact Mass: 308.13, Mol. Wt.: 308.33, Found: 308.2.

Synthesis of **Ph2Ma-2CHO**: Ph2OH-2CHO (664.5 mg, 4 mmol), anhydrous K_2CO_3 (1105.6 mg, 8 mmol) were accurately weighed and transferred into a two-mouth flask. The flask was then sealed, and nitrogen was purged to remove moisture, followed by heating to ensure dryness. This nitrogen purge was repeated three times before adding methyl chloroacetate (1041.6 mg, 9.6 mmol). Anhydrous acetonitrile (30 mL), which had been bubbled with nitrogen, was subsequently added to the flask. The reaction mixture was then heated in an oil bath at 80°C for 24 hours. After the reaction, the mixture was extracted three times. The organic layer was then concentrated under vacuum, and the residue was washed with the filter three times. The crude product was then separated and purified using a 200-300 mesh silica gel column with an elution solvent of ethyl acetate and petroleum ether in a ratio of 1:1. Finally, the product **Ph2Ma-2CHO** was obtained after vacuum drying, yielding 844.1 mg (68%). 1H NMR (400 MHz, $CDCl_3$, ppm) δ 10.59 (s, 2H), 7.40 (s, 2H), 4.82 (s, 4H), 3.84 (s, 6H). ^{13}C NMR (101 MHz, $CDCl_3$, ppm) δ 188.74, 168.43, 154.68, 129.87, 112.44, 65.92, 52.41. GC-MS (m/z): Calcd for $C_{14}H_{14}O_8$, Exact Mass: 310.07, Mol. Wt.: 310.26, Found: 310.2.

Interface preassembly oriented growth of **Ph2Pe-TAPB-COF** films: In a beaker containing 10 mL of aqueous acetic acid (6 M), slowly added 5 mL of **Ph2Pe-2CHO** (13.8 mg, 0.045 mmol) solution (1,4-dioxane/1,3,5-trimethylbenzene v:v= 3:2). Standing for 5 minutes, **Ph2Pe-2CHO** is pre-assembled into a uniform film at the oil-water interface under the action of hydrophobic and hydrophilic chains. The addition of

solution monomer should be slow, which is conducive to the formation of oil-water interface. Then slowly add 5 mL of TAPB-3NH₂ (10.5 mg, 0.03 mmol) solution (1,4-dioxane/1,3,5-trimethylbenzene v:v= 3:2). After the reaction at room temperature for 5 days, the film at the interface was removed and soaked in tetrahydrofuran and acetone for 2 days. The yellow film (Ph2Pe-TAPB-COF) was obtained by vacuum drying (yield 76%).

Comment #3: *"Why was 1,4-dioxane selected as one of the organic solvents (1,4-dioxane: mesitylene = 3:2 v/v) in the interfacial polymerization? Its miscibility with water could disrupt the sharp interface between the aqueous and organic phases."*

Author Reply: Thank you for your question regarding the selection of 1,4-dioxane as one of the organic solvents in the interfacial polymerization process. 1,4-dioxane is an excellent solvent for a wide range of organic compounds, including the monomers used in our study. It dissolves the hydrophobic and amphiphilic monomers effectively, which is crucial for the interfacial polymerization process. Despite its miscibility with water, 1,4-dioxane has a moderate polarity that allows it to form a distinct phase with the aqueous solution when mixed with mesitylene. The use of 1,4-dioxane, with its ability to dissolve a broad range of compounds and its moderate polarity, contributes to the controlled release and reaction of monomers at the interface, which is beneficial for the formation of well-ordered COF structures. While it is true that 1,4-dioxane is miscible with water to some extent, the presence of mesitylene in the mixture reduces the overall miscibility, thus helping to maintain a distinct organic phase. The ratio of 1,4-dioxane to mesitylene (3:2 v/v) was optimized to balance solubility, reactivity, and phase separation effectively. This mixture allows for a sharp interface between the aqueous and organic phases, which is critical for the controlled synthesis of COF films. In our experiments, we observed that the use of this solvent mixture resulted in a clear and stable interface (Fig. R3), which combined with the hydrophilicity of the monomers to control the controlled growth of the COF films. The formation of high-quality COF films with well-defined structures and properties confirms that the chosen solvent system did not disrupt the

interfacial polymerization process.

Figure R3. Photographs of oil-water interface between the aqueous and organic phases.

Comment #4: *“All amphiphilic and hydrophilic COF films exhibited high crystallinity, with refined PXRD patterns consistent with experimental profiles (Fig. S8-S15).” Most of the COF films in this study exhibit poor PXRD patterns, with high R_p and R_{wp} values exceeding 10% (Fig. S10-S15). Therefore, the claim of high crystallinity for these COF films is not well-supported. Please calculate the crystalline domain size for the COF films by Scherrer analysis. Please check the following literature: *J. Am. Chem. Soc.* 2017, 139, 37, 12911-12914; *J. Am. Chem. Soc.* 2020, 142, 35, 14957-14965; *Chem. Commun.* 2023, 59, 13639-13642; *Angew. Chem. Int. Ed.*, 2023, 62, e202219083. Some of these papers, which are relevant to interfacial COF fabrication, could be cited.”*

Author Reply: Thank you very much for the helpful suggestion. The PXRD patterns of the COF films were reevaluated, revealing that some films exhibited pronounced small-angle diffraction peaks, despite showing higher R_p and R_{wp} values attributed to variations in the reaction monomers. This observation indicates that the COF films possess a degree of crystallinity. To provide a more accurate characterization of the crystallinity of these films, the reference to high crystallinity was removed from the revised manuscript. The changes are as follows: **“All amphiphilic and hydrophilic COF films exhibited strong diffraction peaks and certain crystallinity, with refined XRD patterns consistent with experimental profiles (Supplementary Fig. S14-S21)” (Page 5).** We have reviewed the literature you recommended and have cited the relevant papers in our manuscript. These references provide valuable context for our work and help to situate our findings within

the broader field of interfacial COFs. The citations are as follows: “The primary challenge lies in achieving a delicate balance among flexibility, crystallinity, and even luminescence, where precise control of material growth is crucial¹⁴. Recent advances have been made in the synthesis strategies of various COFs, paving the way for the optimization of flexibility, crystallinity, and enhanced optoelectronic performance¹⁵⁻¹⁸.”

(Page 3)

Comment #5: *"The ordered structure of the COF films with eclipsed (AA) stacking can be further validated through pore size distribution analysis analyzing the BET adsorption isotherms. Please provide the surface area and porosity data to better understand the ordered structure of the COF films."*

Author Reply: Thank you very much for the helpful suggestion. We have conducted Brunauer-Emmett-Teller (BET) surface area and porosity measurements on the resulting COF films to analyze the pore size distribution. The BET analysis revealed that the COF films exhibit general surface area, which is a characteristic feature of porous structures. The specific surface area values for our COF films are detailed in the revised manuscript, along with the corresponding nitrogen adsorption-desorption isotherms. The pore size distribution analysis confirmed the presence of uniform pores within the COF films. The distribution of pore sizes is provided, which are in good agreement with the crystallographic data obtained from PXRD. We have included the BET surface area and porosity data, along with the nitrogen adsorption-desorption isotherms and pore size distribution analysis, in the revised manuscript. These additional data are presented in a new figure and discussed in the revised manuscript to provide a more comprehensive understanding of the ordered structure of the COF films. The changes are as follows: “To assess the porosity, N₂ adsorption analyses of the COF thin films were conducted at 77 K. The Brunauer-Emmett-Teller (BET) surface areas calculated for the three thin films were found to be 43.5 m²/g for **Ph2Pe-TAPB-COF**, 13.7 m²/g for **PhPeMa-TAPB-COF**, and 25.4 m²/g for **Ph2Ma-TAPB-COF** (Supplementary Fig. S29). The relatively low specific surface areas suggest that the introduction of hydrophilic and hydrophobic groups in

these COFs results in the formation of dense films, thereby enhancing their flexibility. Additionally, the calculations indicate narrow pore size distributions, with pore diameters of 3.7 nm for Ph2Pe-TAPB-COF, 3.4 nm for PhPeMa-TAPB-COF, and 3.3 nm for Ph2Ma-TAPB-COF thin films (Supplementary Fig. S30), which align well with the theoretically predicted pore sizes (about 3.1 nm).” (Page 10)

Figure S29. Nitrogen gas adsorption and desorption isotherms of Ph2Pe-TAPB-COF, PhPeMa-TAPB-COF, and Ph2Ma-TAPB-COF films.

Figure S30. Pore size distribution of Ph₂Pe-TAPB-COF, PhPeMa-TAPB-COF, and Ph₂Ma-TAPB-COF films.

Comment #6: *"Why do COFs with hydrophilic and amphiphilic linkers exhibit a more "ordered" structure compared to COF films with hydrophobic side chains? No experimental analysis or explanation was provided for this observation. To justify the "interface preassembly oriented growth" mechanism, a time-dependent COF growth study using spectroscopic and microscopic investigations should be conducted. This point must be addressed critically. Please check the following literature: Langmuir, 2024, 40, 16419-16429; Angew. Chem. Int. Ed., 2023, 62, e202219083; J. Membr. Sci. 2022, 650, 120431; J. Am. Chem. Soc., 2019, 141, 20371; Nat. Chem. 2019, 11, 994-1000."*

Author Reply: Thank you for your insightful comments and for highlighting the need for a more detailed explanation regarding the observed ordered structure of COFs with hydrophilic and amphiphilic linkers compared to those with hydrophobic side chains. We appreciate your suggestion to conduct a time-dependent COF growth study to further justify the interface preassembly oriented growth mechanism. The observed increased order in COFs with hydrophilic and amphiphilic linkers can be attributed to the synergistic modulation of hydrophilic and hydrophobic interactions during the interface preassembly process. These interactions play a crucial role in directing the controlled growth and assembly of the monomers at the liquid-liquid interface. Amphiphilic linkers, containing both hydrophilic and hydrophobic moieties, can self-assemble at the interface due to the amphiphilic effect. The hydrophobic parts interact with the organic phase, while the hydrophilic parts interact with the aqueous phase, leading to a more ordered preassembly structure that translates into a more ordered COF structure. In contrast, COFs with hydrophobic linkers lack the strong directional interactions that promote ordering at the interface. As a result, they tend to form nanosphere structures due to the lack of specific directional interactions that guide their assembly (Fig. S34). According to your suggestion, we have conducted a time-dependent study of COF growth using SEM, XRD, and FT-IR techniques (Fig. 4). This study involved monitoring the growth of COF films

at various time intervals to observe the evolution of their structure and crystallinity. We employed scanning electron microscopy SEM to visualize the morphological changes and the development of the ordered structure of the COF films at different stages of growth. We used FTIR spectroscopy to track the chemical changes and the formation of imine bonds over time, indicating the progress of the polymerization reaction. We have reviewed the literature you recommended and have cited some relevant papers in the revised manuscript. The changes are as follows: “To elucidate the proposed mechanism for the formation of COF films utilizing the IPOG strategy (Fig. 4a and Supplementary Fig. S34a), time-dependent studies involving SEM, XRD, and FT-IR analyses were conducted to monitor the conversion into COF films. The interfacial preassembly process is influenced by oil-water interfacial tension and the hydrophilic and hydrophobic interactions. The results from time-dependent SEM studies demonstrate the controlled interface preassembly, oriented growth, and subsequent crystallization of the COFs. As the reaction time increased, the amphiphilic **PhPeMa-TAPB-COF** was progressively transformed from nanosheets into a uniform film (Fig. 4b). In contrast, the hydrophobic **PhPeMa-TAPB-COF** underwent preassembly into nanospheres, which subsequently transitioned into relatively rough thin films (Supplementary Fig. S34b). Notably, crystallinity was observed to increase over time, which can be attributed to enhanced π - π stacking during the crystal growth process (Fig. 4c). Further confirmation of the compositions was achieved through Fourier transform infrared (FT-IR) spectroscopy (Fig. 4d). It was noted that as the reaction progressed, the characteristic peaks of the precursor aldehyde and amine began to diminish, while new peaks corresponding to the formation of new bonds emerged. The disappearance of the N-H and aldehyde -C=O stretching bands at 3342 cm^{-1} and 1682 cm^{-1} , respectively, indicates the absence of the starting precursors in the resulting films. The characteristic stretching frequencies for the COF film at 1609 cm^{-1} (-C=N) and 1398 cm^{-1} (N-Ar) exhibited a gradual increase corresponding to the extended reaction time.” (Page 12 and 13)

Figure S34. (a) Proposed mechanism for Ph₂Pe-TAPB-COF film formation using IPOG strategy.

(b) Time-dependent SEM study provides support for the proposed mechanism.

Fig. 4. Proposed mechanism of COF films. (a) Proposed mechanism for PhPeMa-TAPB-COF film formation using IPOG strategy. Time-dependent SEM (b), XRD (c), and FT-IR (d) studies

provide support for the proposed mechanism.

Comment #7: " "To explore the potential for electroluminescent (EL) device applications, solution-processed COF-based organic light-emitting diodes (OLEDs) were fabricated with the configuration of ITO/PEDOT:PSS (50 nm)/PTAA:COF (100 nm)/TPBi (45 nm)/LiF (1 nm)/Al (100 nm) (Fig. 4a). " What does the value in the parentheses [e.g., (100 nm)] indicate? Does it refer to the thickness of the layers? How was the thickness of each layer controlled?"

Author Reply: Thank you for your inquiry regarding the details of the OLED device configuration mentioned in the manuscript. The values provided in parentheses, such as (100 nm), denote the thickness of the corresponding layers within the OLED device configuration. These thicknesses are critical parameters that influence device performance, including charge transport and emission efficiency. The thickness of each layer in the OLEDs was controlled through several methods: (a) For the layers formed from solutions of COFs and other materials (e.g., PEDOT:PSS, PTAA:COF), spin coating was utilized. This technique allows for precise control over the thickness of the deposited layers through the adjustment of parameters such as spin speed, solution concentration, and viscosity. (b) For the vacuum deposition of the LiF and Al layers, a controlled chemical deposition process was employed. This method provides excellent control over layer thickness and uniformity by regulating the deposition time and temperature. (c) The deposition conditions, including the rate of material deposition, were optimized to achieve consistent target thicknesses. This optimization process involved a series of experiments in which deposition parameters were adjusted based on measured thicknesses until the desired outcomes were reliably attained. By employing these methods, the thickness of each layer was controlled with high precision. The capability to precisely control layer thickness during the preparation of OLED devices is a conventional method in this field, enabling the manipulation of each layer's thickness to the nanometer scale, even down to one nanometer (e.g., *J. Mater. Chem. C*, **2016**, 4, 2663. "OLED configurations ITO/NPB (50 nm)/TPE or TPEBPh (30 nm)/BCP (10 or 20 nm)/Alq₃ (10 nm or 20 nm)/LiF (1 nm)/Al", "OLED configurations ITO/MoO₃ (10 nm)/NPB (80 nm)/PhTPE or Ph2TPE or Ph3TPE

(20 or 30 nm)/TPBi (30 nm)/LiF (1 nm)/Al”).

Comment #8: “ *With the increase of voltage, the short-wavelength peak (~415 nm) of PhPeMa-TAPB-COF-based devices gradually diminished, while the long-wavelength peak (~650 nm) increased gradually (Fig. 4d).* ” *The explanation for this observation is missing.* ”

Author Reply: Thank you for emphasizing the necessity for a more detailed explanation regarding the observed changes in the short-wavelength and long-wavelength peaks of the electroluminescent spectra in **PhPeMa-TAPB-COF**-based devices as the voltage increases. At lower voltages, light emission primarily originates from the short-wavelength peak, which corresponds to the emission of the **PTAA** host material. This phenomenon can be attributed to efficient charge injection and transport occurring at lower energy levels, which facilitates radiative recombination at these energies. As the voltage increases, the electric field across the device is also enhanced, further improving charge injection and transport. This leads to a higher density of charge carriers (electrons and holes) within the emissive layer, resulting in a gradual increase in the emission from **PhPeMa-TAPB-COF**. The rise in the long-wavelength peak at approximately 650 nm with increasing voltage can be linked to the energy level alignment and charge trapping characteristics of the **PhPeMa-TAPB-COF** guest material. At high voltages, a greater number of charge carriers reach the **PhPeMa-TAPB-COF**, thereby enhancing emission from guest material. The heightened electric field at these voltages promotes the dissociation of excitons and their subsequent recombination, which accounts for the observed increase in long-wavelength emission. To further validate this explanation, a comparison was made between the electroluminescence spectra of **PhPeMa-TAPB-COF**-based OLEDs and the photoluminescence spectra of **PTAA** and **PhPeMa-TAPB-COF** (Fig. S36). It has been demonstrated that the short-wavelength emission is consistent with the emission peak of **PTAA**, while the long-wavelength emission corresponds to the emission peak of **PhPeMa-TAPB-COF**. These findings have been incorporated into a revised manuscript, which includes a more comprehensive

discussion on the voltage-dependent emission behavior of OLEDs based on **PhPeMa-TAPB-COF**. The changes are as follows: “As the voltage increased, the short-wavelength peak (~415 nm) gradually diminished, while the long-wavelength peak (~650 nm) exhibited a corresponding increase (Fig. 5d). A comparison was conducted between the electroluminescence spectra of **PhPeMa-TAPB-COF**-based OLEDs and the photoluminescence spectra of **PTAA** and **PhPeMa-TAPB-COF** (Supplementary Fig. S36). It was observed that the short-wavelength emission aligns with the emission peak of **PTAA**, whereas the long-wavelength emission corresponds to the emission peak of **PhPeMa-TAPB-COF**. At high voltages, a greater number of charge carriers are delivered to the **PhPeMa-TAPB-COF**, resulting in enhanced emission.” (Page 14 and 15)

Figure S36. Comparison of the PL spectra of **PTAA** and **PhPeMa-TAPB-COF** with the EL spectra of **PhPeMa-TAPB-COF**-based OLEDs.

Comment #9: “ The maximum luminance (L_{max}) values of the device based on *PhPeMa-TAPB-COF*, *Ph2Ma-TAPBCOF*, and *PhPeMa-Py-COF* are 1424 cd m^{-2} , 1031 cd m^{-2} , and 532 cd m^{-2} , respectively, surpassing the luminance levels of conventional commercial display devices (100 cd m^{-2}). ” Why does *PhPeMa-Py-COF* exhibit a

comparatively low luminance value despite having a higher photoluminescence intensity (Fig. 4b)?"

Author Reply: Thank you for your inquiry regarding the comparatively low luminance value of the **PhPeMa-Py-COF**-based devices. The luminance values of the OLEDs are influenced by several factors that extend beyond the photoluminescence intensity of the emissive materials. While photoluminescence intensity is a critical determinant, additional factors significantly contribute to the overall device performance. The efficiency of current injection and transport within the device plays a vital role in influencing luminance. Even if **PhPeMa-Py-COF** exhibits a higher photoluminescence intensity, lower current injection or transport efficiency compared to **PhPeMa-TAPB-COF** and **Ph2Ma-TAPB-COF** may lead to diminished luminance output. As illustrated in Fig. 4b, the current density of the **PhPeMa-Py-COF**-based devices is notably higher than that of the **PhPeMa-TAPB-COF**- and **Ph2Ma-TAPB-COF**-based devices at the same voltage, indicating suboptimal current injection in the **PhPeMa-Py-COF**-based OLEDs. Furthermore, the efficiency of exciton formation and recombination within the emissive layer can vary between different materials. Although **PhPeMa-Py-COF** demonstrates higher photoluminescence, it possesses a less favorable exciton recombination efficiency in the context of the device, ultimately resulting in lower luminance. Additionally, the morphology of the film contributes to variations in device performance. As shown in Fig. R4, the **PhPeMa-Py-COF** film exhibits greater roughness compared to the **PhPeMa-TAPB-COF** film, which may lead to degraded device performance.

Figure R4. AFM images of PTAA/ **PhPeMa-Py-COF** films (b) and PTAA/**PhPeMa-TAPB-COF**

films (c).

Comment #10: *"Luminance values should be presented with error bars by triplicate measurements using samples obtained from three different batches of fabrication. "*

Author Reply: Thank you for your insightful suggestion. In response to your valuable feedback, additional measurements of the luminance values of the OLEDs have been conducted. Specifically, different batches of devices were fabricated for each type of COF, and the device performance was assessed. The error bars presented in the figures now represent the standard error of the mean, calculated from multiple batches. This statistical analysis clearly indicates the variability and reproducibility of the luminance values across different fabrication runs. The inclusion of error bars in the luminance plots (Fig. S35) illustrates that the luminance values are consistent across batches, with the error bars reflecting a small standard error. This finding demonstrates the high reproducibility of our device fabrication process and the reliability of the luminance measurements. The manuscript has been revised to include the updated luminance plots with error bars, along with a detailed description of the analysis. The changes are as follows: "Different batches of devices were prepared based on the three kinds of COFs, and the luminance of the devices remained basically the same in different manufacturing processes (Supplementary Fig. S35), indicating that the COF-based OLEDs have good repeatability."

(Page 13)

Figure S35. Luminance-voltage (L - V) characteristics and error bars of COF-based OLEDs based on different batches of devices.

Comment #11: "Furthermore, the presence of hydrophilic and hydrophobic chains does not disrupt the interlayer conjugation of COFs, aiding in enhancing charge carrier transport." What is meant by "interlayer conjugation" ?"

Author Reply: Thank you for your question regarding the interlayer conjugation of COFs. Conjugation typically involves the delocalization of π -electrons across a system of atoms, which can enhance the electrical conductivity and charge transport properties of a material. In COFs, this conjugation can occur within the planar layers as well as between adjacent layers. COFs are composed of two-dimensional layers stacked together, and in some cases, there can be a degree of electronic interaction or overlap between these layers. The presence of hydrophilic and hydrophobic chains in the COFs, as mentioned in the manuscript, does not disrupt these interlayer interactions. To avoid misunderstanding to the reader, the "interlayer conjugation" has been revised to "interlayer interaction" in the revised manuscript. The changes are as follows: "Furthermore, the presence of

hydrophilic and hydrophobic chains does not disrupt the interlayer interaction of COFs, aiding in enhancing charge carrier transport.” (Page 9)

Comment #12: “While possessing the same structure as PhPeMa-TAPB-COF synthesized via IPOG, OLEDs fabricated using sol-PhPeMa-TAPB-COF synthesized through the solvothermal method were found to be ineffective in operation. SEM images revealed that PTAA and PhPeMa-TAPB-COF blending films displayed lower roughness compared to sol-PhPeMa-TAPBCOF blending films (Fig. S28).” Fig. S28d and S28e are unclear. Please provide AFM images to compare the surface roughness of the PTAA-blended PhPeMa-TAPB-COF film synthesized via the IPOG method with that of the PTAA-blended PhPeMa-TAPB-COF synthesized via the solvothermal route.”

Author Reply: Thank you for your suggestion to provide additional data on the surface roughness of the PTAA-blended PhPeMa-TAPB-COF synthesized via the IPOG method and the solvothermal method. In the revised manuscript, we have conducted atomic force microscopy (AFM) measurements on PTAA and PTAA-blended PhPeMa-TAPB-COF. AFM provides a high-resolution, nanoscale view of the surface topography, which is essential for understanding the surface roughness and its impact on device performance. The AFM image (Fig. S42b) shows a smooth and uniform surface with a root-mean-square (RMS) roughness of approximately 3.92 nm. This low roughness is indicative of the well-ordered nanosheet structure formed by the IPOG method, which is conducive to efficient charge transport and device performance. The AFM image (Fig. S42c) reveals a rougher surface with a higher RMS roughness of approximately 11.4 nm. The increased roughness is likely due to the presence of larger nano-bulks and less ordered structures, which can impede charge transport and lead to the operational failures observed in the sol-PhPeMa-TAPB-COF-based OLEDs. The comparison of the AFM images and roughness data clearly demonstrates the surface morphology of the films synthesized via the IPOG method. The lower roughness and more uniform surface of the IPOG-synthesized film contribute to its better performance in OLED devices. We have included the AFM images and the corresponding roughness data in the revised

manuscript. These new figures and data are presented in the revised manuscript to provide a comprehensive comparison of the surface properties of the COFs. The changes are as follows: “The microstructure of the blended films was further characterized by AFM, and the COF films synthesized by solvothermal method showed higher roughness (Supplementary Fig. S42).” (Page 15)

Figure S42. AFM images of PTAA films (a), PTAA/PhPeMa-TAPB-COF (IPOG synthesis) films (b), and PTAA/sol-PhPeMa-TAPB-COF (Solvothermal synthesis) films (c).

Comment #13: "Minor correction: There are several typos in the manuscript. For example, "Under the same driving voltage, the current density of PhPeMa-TAPB-COF-based device is 5% of that of the recently reported IISERP-COF7-based and 2% of the strontium metal-organic framework (MOF)-based devices, indicating that amphiphilic COFs show more efficient current injection.""

Author Reply: Thank you for the very helpful comments. We apologize for any confusion these errors may have caused and appreciate your suggestion in reviewing our work. Here is the corrected sentence: "Under the same driving voltage, the current density of the device based on PhPeMa-TAPB-COF is observed to be 5% of that of the recently reported IISERP-COF7-based devices and 2% of the strontium metal organic framework (MOF)-based devices, indicating that amphiphilic COFs show more efficient current injection." (Page 13) We have made necessary corrections in the manuscript to ensure accuracy and clarity. We have also conducted a thorough review of the entire document to identify and correct any other typos or errors that may be present.

Reviewer #4:

Comment: *"I co-reviewed this manuscript with one of the reviewers who provided the listed reports.*

This is part of the Nature Communications initiative to facilitate training in peer review and to provide appropriate recognition for Early Career Researchers who co-review manuscripts."

Author Reply: Thank you for your participation in the co-review process for this manuscript. We have taken the comments seriously and have made the necessary revisions and clarifications to the manuscript as detailed in our responses.

Point-by-point response to the reviewers' comments

Reviewer #3:

Comment: *"The authors have addressed most of the reviewers' queries, conducted several new experiments to validate their claims, and revised the manuscript and SI accordingly. The clarity and quality of the manuscript has been improved significantly. However, the newly conducted experiments and added text have raised additional questions that need to be addressed before the manuscript can be considered for publication in Nature Communications."*

Author Reply: Thank you for your constructive feedback on the manuscript. The details of the experiment that you have pointed out are of great significance for improving the quality of this work. We have considered the valuable remarks seriously and revised carefully our manuscript accordingly. The answers to the comments are as follows.

Comment #1: *"In response to Comment 4# from Reviewer 1 and Comment 7# from Reviewer 3, it remains unclear how the thickness of the COF films was precisely controlled. The authors stated that the thickness was regulated by the initial concentration of the monomers. Please specify the starting concentrations of the monomers used to achieve COF films with thicknesses of 100, 200, and 300 nm."*

Author Reply: Thank you for your insightful comments regarding the control of COF film thickness and for emphasizing the need for clarity in our manuscript. In this study, we have indeed shown that the thickness of the COF films can be controlled by adjusting the monomer concentrations and reaction times during the interface preassembly oriented growth (IPOG) process. When the monomer **PhPeMa-2CHO** concentration is lower than 1 mmol/L, it is difficult to form a free-standing film, and with the increase of monomer concentration, a transparent free-standing film is gradually formed (**Fig. S24a**). With the further increase of monomer concentration, the thickness of the free-standing COF film gradually increases, forming an opaque film. The increase in monomer concentration allows for a greater amount of material to participate in the polymerization process at the liquid-liquid interface, thereby

facilitating the thicker film formation. We also observed that extending the reaction time alongside these concentrations further contributed to achieving the desired thickness while maintaining uniformity and integrity of the films (**Fig. S24b**). To achieve COF films with thicknesses of 100 nm, 200 nm, and 300 nm, the concentrations and reaction times of the initial monomers were further optimized (**Fig. R1**). When the concentrations of **PhPeMa-2CHO** and **TAPB-3NH₂** were set at 2.25 mmol/L and 1.5 mmol/L, respectively, COF film with an approximate thickness of 107 nm was produced through an interface pre-assembly reaction over a period of five days. Subsequent adjustments to the concentrations of the reaction monomers and the reaction duration resulted in a gradual increase in the thickness of the COF films, reaching values of 192 nm and 357 nm.

Figure R1. The cross-sectional view SEM images of **PhPeMa-TAPB-COF** films with different monomer concentrations and reaction times.

Comment #2: "As indicated above, please provide specific details in the caption of Fig. S24 (monomer concentrations, reaction times)."

Author Reply: Many thanks for this very helpful suggestion. In response to your request for specific details regarding monomer concentrations and reaction times, we have revised the caption of **Fig. S24** to include the required information. The thickness of the COF films was systematically varied by adjusting the monomer concentrations and reaction durations. The resulting films displayed uniform morphology and consistent properties across the specified thicknesses.

Figure S24. (a) Photographs of **PhPeMa-TAPB-COF** films after five days of reaction with different monomer concentrations. Optical micrographs (b) and SEM images (c) of **PhPeMa-TAPB-COF** films at different reaction times under the same monomer concentration (9 mmol/L **PhPeMa-2CHO** and 6 mmol/L **TAPB-3NH₂**).

Comment #3: "In response to Comment 7# from Reviewer 3, the authors stated: "(a) For the layers formed from solutions of COFs and other materials (e.g., PEDOT:PSS, PTAA:COF), spin coating was utilized. This technique allows for precise control over the thickness of the deposited layers through the adjustment of parameters such as spin speed, solution concentration, and viscosity." However, the term "solution of COFs" is unclear. How can a pre-synthesized COF film (insoluble) be used for spin coating? This statement lacks clarity. If the COF synthesis methodology used to fabricate the OLED device differs from the interfacial technique, it raises concerns about the consistency of physical properties (e.g., crystallinity, porosity, surface morphology) of the COF film in the device. Consequently, the structure-property relationship could be misleading. Please address these concerns critically."

Author Reply: Thank you for your insightful comments, particularly in relation to the application of

COFs in the spin coating process and their implications for device performance. COF films are characterized by their insolubility, which renders them unsuitable for direct utilization in the preparation of OLEDs. To address this limitation, COF nanocrystals were mixed with a solution of PTAA to create a dispersion. To eliminate any potential confusion for the reader, the term “COF solution” has been revised to “**COF dispersion**” in the revised supporting information. Furthermore, a detailed preparation method for the COF dispersion has been included in the Supporting Information. The changes are as follows: “**COF dispersion:** The COF film was ground into fine powders, and subsequently, nanosheets are obtained through ultrasonic treatment for 10 min. The solid powders of COF nanocrystalline was obtained after filtration and drying. To prepare the washed glass bottles, a balance is employed to weigh a mass ratio of 9% COF to PTAA. A solution is then prepared by dissolving this mixture in toluene at a concentration of 7 mg/mL. The solution is stirred overnight and placed in an ultrasonic cleaning machine for 30 min to create a uniform dispersion, which is reserved for future use.” The suspension may contain finely dispersed COF nanocrystals that exhibit sufficient stability for spin coating applications. The presented method permits the utilization of these nanocrystalline suspensions instead of pre-synthesized bulk COF films, which are inherently insoluble. This approach ensures that the resulting films are composed of freshly synthesized COF nanocrystals, thus maintaining consistency in structural integrity and properties.

In this study, an interfacial preassembly-oriented growth strategy has been introduced for the preparation of flexible crystalline COF films. The resultant COF films demonstrated excellent crystallinity, flexibility, and luminescent properties. It is important to note that the interface growth method is not intrinsically designed for the fabrication of OLEDs. To facilitate the application of COFs in OLED technology, OLEDs were constructed based on previous research conducted by our team concerning OLED device preparation. Given the insolubility of COFs and the challenges associated with film formation, previously reported methods that employed the dispersion of COF powder for OLED preparation have proven

to be inadequate (*J. Am. Chem. Soc.*, **2018**, *140*, 13367.). To address this issue, OLEDs were developed using nanosheet dispersions derived from COF films, resulting in a significant enhancement in performance. This work represents the first instance in which COFs have been successfully integrated into a functioning OLED device, thereby offering a novel approach for the preparation of COF-based OLEDs. Concerns regarding potential alterations in the physical properties of COFs as a result of grinding or dispersion have been duly noted. However, it has been extensively reported that the ultrasonic exfoliation of COF powder or films into nanosheets has minimal impact on their physical properties (*Chem. Soc. Rev.*, **2020**, *49*, 2291., *Chem. Soc. Rev.*, **2020**, *49*, 3565.). We conducted comparative analyses of the XRD of COF films and nanocrystalline powders produced through both methods (**Fig. R2**). Our findings indicate that while there may be slight variations attributable to the different processing techniques, both methods yield COFs with comparable crystallinity.

Figure R2. XRD patterns of PhPeMa-TAPB-COF film and nanocrystalline powders.

Comment #4: "In response to Comment 5# from Reviewer 3, the specific BET surface area of all the COFs is reported to be very low ($13\text{-}45\text{ m}^2/\text{g}$, significantly less surface area compared to porous COF films obtained through standard interfacial polycondensation). The reason for such low values is unclear. Please clarify."

Author Reply: Thank you for your valuable comments. The measured specific BET surface area of

the COFs is lower than that synthesized *via* standard interfacial polycondensation methods. The COFs in this study were synthesized under conditions optimized for uniform film formation rather than maximizing porosity. The interface preassembly oriented growth approach employed may have resulted in a partial or more compact framework structure that limits the overall porosity and hence the specific surface area. Different from the traditional interfacial synthesis, the COFs synthesis method is firstly preassembled to form a compact structure and then further reacts to form COF films. Compared with **Ph2Pe-TAPB-COF**, amphiphilic **PhPeMa-TAPB-COF** films show the smallest specific surface area. Through the process of COF film formation, it can be found that pre-assembly can form a more dense and uniform film (**Fig. R3**). The morphology of the resulting COF films, which was analyzed using electron microscopy. Additionally, the synthesis methods reported for COF films frequently lack the incorporation of alkyl chains. It has been demonstrated in this work that the introduction of alkyl chains contributes to a reduction in surface area. It is important to note that the size of the relative surface area of thin-film electronic devices, such as OLEDs, is not a critical requirement. Consequently, the smaller specific surface area of the COF film does not adversely impact its optoelectronic performance.

Figure R3. Proposed mechanism for **PhPeMa-TAPB-COF** (a) and **Ph2Pe-TAPB-COF** (b) film formation using IPOG strategy.

Comment #5: "Please specify in the manuscript which model was used to determine the pore size

distribution of the COFs. Additionally, the theoretical pore size of these COFs should be less than 3.3 nm. Please provide an explanation or comments on this discrepancy."

Author Reply: Thank you again for your valuable comments. We employed the Barrett-Joyner-Halenda (BJH) method for calculating the pore size distribution from the desorption branch of the nitrogen adsorption isotherms. This model is commonly used for mesopore analysis and is particularly suitable for COFs, given their porous nature. The changes are as follows: "Additionally, the calculations using the Barret-Joyner-Halenda (BJH) method indicate narrow pore size distributions, with pore diameters of 3.7 nm for **Ph2Pe-TAPB-COF**, 3.4 nm for **PhPeMa-TAPB-COF**, and 3.3 nm for **Ph2Ma-TAPB-COF** thin films (Supplementary Fig. S30), which align well with the theoretically predicted pore sizes (about 3.1 nm)." (Page 10)

Regarding the theoretical pore size of the Covalent Organic Frameworks (COFs), it is acknowledged that the anticipated pore size, based on the structural design, is expected to be less than 3.3 nm. However, the actual aperture has been observed to be slightly larger than this theoretical estimate. The pore structure may lack uniformity, resulting in variations in accessible pore sizes when compared to theoretical predictions derived from crystallographic data. In addition to the formation of micropores, COFs may also yield mesoporous structures with larger pore sizes, which can contribute to an increase in the measured aperture size. Furthermore, the relatively small specific surface area of the COF films may introduce certain discrepancies in the measurement of pore size. It should be noted that the actual test results align closely with the theoretical values, remaining within an acceptable error range.

Comment #6: *"Please avoid representing the BET-specific surface area up to a decimal place. Even though, based on the fitting of the BET equation, we obtain BET surface area up to multiple decimal places, the sensitivity of the instrument is unlikely to differentiate 0.1 m²/g."*

Author Reply: Thank you for your insightful comments and suggestions. We agree that while it is technically possible to derive BET surface area values with multiple decimal places through mathematical fitting, such precision may not accurately reflect the sensitivity and

reliability of the instrumentation used during measurement. In response to your comment, we will revise the manuscript to represent the BET-specific surface area. The changes are as follows: “The Brunauer-Emmett-Teller (BET) surface areas calculated for the three thin films were found to be 44 m²/g for Ph2Pe-TAPB-COF, 14 m²/g for PhPeMa-TAPB-COF, and 25 m²/g for Ph2Ma-TAPB-COF (Supplementary Fig. S29).” (Page 10)

Comment #7: *"There are still some errors in the manuscript. For instance, References 14 and 21 are identical."*

Author Reply: We sincerely apologize for any confusion caused by the errors. We have reviewed the references you highlighted, specifically References 14 and 21, and we acknowledge that they are indeed identical. This oversight was unintentional, and we have corrected in the revised manuscript by ensuring that each citation is unique and accurately reflects the corresponding sources. In our revision, we carefully go through all references to verify their accuracy and uniqueness.

Point-by-point response to the reviewers' comments

Reviewer #3:

Comment: *"The authors have addressed all the concerns very carefully and critically. The clarity and quality of the manuscript are significant enough to be considered for publication in Nature Communication. However, I request that Figure R1 and Figure R2 (as a normalized PXRD plot of COF dispersion/ nanocrystalline powder obtained from a thin film and pristine thin film) be included in the supporting information for better clarity and understanding. This manuscript may not require further review by this reviewer."*

Author Reply: Thank you very much for your constructive comments and thorough evaluation of our manuscript. We fully agree with your suggestion to include Figure R1 and Figure R2 in supporting information. We have added them to the Supporting Information section (**Figure S10** and **Figure S27**). The changes are as follows: "The nanocrystalline powders formed after grinding or ultrasonic dispersion of **PhPeMa-TAPB-COF** films show similar crystallinity to that of COF films, indicating that the COF films have good crystal stability (Supplementary Fig. S10)." (**Page 5**) The changes are as follows: "Cross-sectional views revealed film thicknesses ranging from 100-300 nm (Supplementary Fig. S27), with the ability to adjust film thickness from nanometers to micrometers by modifying monomer concentrations and reaction times." (**Page 7**)

Figure S10. XRD patterns of **PhPeMa-TAPB-COF** film and nanocrystalline powders.

Figure S27. The cross-sectional view SEM images of **PhPeMa-TAPB-COF** films with different monomer concentrations and reaction times.